# Impact of metabolic syndrome and its components on bone remodeling in adolescents

**Valéria Nóbrega da Silva[1], Tamara Beres Lederer Goldberg[1]\*, Carla Cristiane Silva[2], Cilmery Suemi Kurokawa[3], Luciana Nunes Mosca Fiorelli[1], Anapaula da Conceição Bisi Rizzo[1], José Eduardo Corrente[4]**

1 Department of Pediatrics, Postgraduate Program in Tocogynecology, Discipline of Adolescent Medicine, Botucatu Medical School, UNESP, São Paulo State University, Botucatu, SP, Brazil, 2 Department of Physical Education, University of North Paraná, Jacarezinho, Paraná, Brazil, 3 Department of Pediatrics, Clinical and Experimental Pediatric Research Center, Botucatu Medical School, UNESP, São Paulo State University, Botucatu, SP, Brazil, 4 Department of Statistics, Biosciences, UNESP, São Paulo State University, Botucatu, SP, Brazil

\* tamara.goldberg@unesp.br

## Abstract

### Introduction

Osteoporosis and metabolic syndrome (MetS) are diseases that have serious public health consequences, reducing the quality of life of patients and increasing morbidity and mortality, with substantial healthcare expenditures.

### Objective

To evaluate the impact of MetS on bone mineral density (BMD) and biochemical markers of bone formation and resorption in adolescents with excess weight.

### Method

A descriptive and analytical cross-sectional study was performed that evaluated 271 adolescents of both sexes (10 to 16 years). From the total sample, 42 adolescents with excess weight and the presence of MetS (14%) were selected. A further 42 adolescents with excess weight and without MetS were chosen, matched for chronological age, bone age, and pubertal developmental criteria to those with MetS, for each sex. Anthropometric measurements, blood pressure collection, and biochemical tests were performed in all adolescents, as well as evaluation of BMD and the bone biomarkers osteocalcin (OC), bone alkaline phosphatase (BAP), and carboxy-terminal telopeptide (S-CTx).

### Results

The adolescents with excess weight and MetS exhibited significantly lower transformed BMD and concentrations of BAP, OC, and S-CTx compared to the matched group, except for OC in boys. A negative and significant correlation was observed between total body

**Funding:** Supported by FAPESP (Fundação de Amparo à Pesquisa do Estado de São Paulo) – Grants (07/07731-0, 2011/05991-0 and 2015/04040-2) and the Pro-Rector for Research of UNESP.

**Competing interests:** The authors have declared that no competing interests exist.

BMD and BAP (r = -0.55568; p = 0.005), OC (r = -0.81760; p = < .000), and S-CTx (r = -0.53838; p = 0.011) in girls.

## Conclusion

Metabolic syndrome may be associated with reduced bone mineral density and biochemical markers of bone formation and resorption in adolescents with excess weight.

## 1. Introduction

Osteoporosis and metabolic syndrome (MetS) are diseases that have serious public health consequences, reducing the quality of life of patients and increasing morbidity and mortality, with substantial healthcare expenditures [1]. The occurrence of MetS in adolescents is associated with an increased risk of cardiovascular diseases, psychosocial problems, abnormal glucose metabolism, nonalcoholic fatty liver disease, polycystic ovary syndrome, obstructive sleep apnea, mental disorders, orthopedic complications, and motor development disorders [2]. In addition to these conditions, although a possible relationship has been demonstrated between MetS and bone mass, the number of studies addressing this topic in adolescents is still small [3]. Despite the paucity of evidence, the results of some studies reveal that MetS is negatively correlated with bone mass and that some components of MetS are negatively correlated with BMD in adolescents [4–8].

The period of pubertal development during adolescence is important for the acquisition of bone mineral content. Especially in this phase of life, which is dominated by physical growth, there is imbalance between bone formation and resorption, and the biochemical formation markers are correlated with the rate of growth [9]. Under normal physiological conditions, bone resorption and formation are dependent phenomena and the predominance of one over the other may lead to the gain or loss of bone mass [10].

The relationship of bone mass with the occurrence of MetS in children and adolescents has been a subject of interest to some researchers [3–6, 11], since impairment in bone acquisition during this period can lead to "suboptimal" peak bone mass and increase the risk of developing osteopenia/osteoporosis and fractures in old age [7]. A recent systematic review performed a critical analysis of studies that focused on the period of adolescence, evaluating the influence of MetS and its components on bone mineral density (BMD) in this age group [3].

In adolescents, the relationship between MetS and markers of bone formation and resorption is not fully known. However, some lines of evidence indicate a negative correlation between serum levels of the bone marker osteocalcin and metabolic risk factors in obese children and adolescents [12]. In view of the results already published [3, 5, 6], the objective of the current study was to evaluate the impact of MetS on BMD, as well as the impact of MetS and each of its components on biochemical markers of bone formation and resorption in adolescents.

## 2. Material and methods

### 2.1 Study design and participants

This was a descriptive and analytical cross-sectional study involving a sample of 271 adolescents with excess weight [5]. For methodological reasons, the final sample consisted of only 84 adolescents because of the need to analyze two matched groups with and without MetS. The

groups were matched for chronological age, bone age, and pubertal stage for each sex. Our sample of 84 adolescents did not differ from that of the 271 initially included subjects. All adolescents of both sexes with and without MetS and with excess weight were recruited on the occasion of their first visit to the Adolescent Unit of the Botucatu Medical School University Hospital (SP, Brazil), between 2011 and 2013.

The project was approved by the Ethics Committee of the Botucatu Medical School (FMB-UNESP) (Approval N°. 1.399.709 /2016). Signed informed consent form was obtained from all participating adolescents and their parents or guardians.

The inclusion criteria were as follows: 10 to 16 years of age, overweight (BMI ≥85th percentile for sex and age), and between the 5th and 95th age- and sex-appropriate height percentiles [13]. Adolescents were excluded from this study for the following reasons: weight exceeding 100 kg (densitometry is not feasible for these subjects), a history of prematurity, a history of fracture in the previous two years, prolonged therapy with corticosteroids, use of calcium and/or iron supplements in the 12 months preceding this research, presence of chronic diseases, use of drugs that negatively affect bone metabolism such as medroxyprogesterone acetate, anticonvulsants, or antacids containing aluminum, consumption of an exclusive high-fiber diet containing above age-appropriate recommendations (>26 g/day for female adolescents or >31 g/day for male adolescents (9–13 years) or >38 g/day for male and female adolescents 14–16 years) [14], a vegan or vegetarian diet, caffeine intake >300 mg/day (>3 cups of coffee per day) [15] consumption of more than 500 mL/day of soft drinks [16, 17], and lack of daily consumption of dairy products. Furthermore, the adolescents did not smoke or drink alcohol, were not engaged in any extracurricular sporting activities, and participated in physical education classes in their schools for no more than 2 h/week. Other exclusion criteria were the presence of metabolic, endocrine, or genetic diseases (as verified by a history of current disorders, general and specific physical examinations, laboratory or radiological procedures), any changes in menstrual cycle (oligo/amenorrhea, clinical or biochemical hyperandrogenism) that could suggest the presence of polycystic ovary syndrome, use of combined oral contraceptives, and pregnancy in girls. These exclusion criteria were established in order to prevent other events or situations described in the literature from interfering with the increase in bone mass in these adolescents. If adolescents presented any mental alterations or used medication to treat mental disorders they were not included in the study.

Adolescents of both sexes were excluded if they failed to attend all scheduled appointments for anthropometric measurements or blood sampling or if they did not follow the established sample collection procedures; e.g., if they did not fast for at least 10 h prior to collection.

## 2.2 Anthropometric assessment and blood pressure measurement

Anthropometric measures of weight (kg) and height (m) were obtained as recommended by the National Health and Nutrition Examination Survey [18], from which the body mass index was calculated (BMI, kg/m$^2$), utilized to assess nutrition status. The BMI percentiles, weight for age, height for age, and BMI for age Z-scores were obtained with Epi Info 3.5.1 Software, which uses sex and age growth curves as references [18]. Adolescents with a BMI between the 85th and 95th percentile were classified as overweight, between the 95th and 99th percentile as obese [19], and above the 99th percentile as extremely obese [20]. Waist circumference (cm) was obtained at the midpoint between the last rib and the iliac crest at the end of a regular expiration [21]. Blood pressure was measured with the adolescent in a sitting position after a 5-min rest. The mean of three measurements with 3-min intervals was used. Measurements were performed on the right arm by auscultation, using a calibrated mercury sphygmomanometer and a cuff appropriate for the arm circumference [22].

### 2.3 Evaluation of sexual maturation and skeletal maturation stages

Sexual maturity was evaluated by an experienced physician through visual breast evaluation (females) or genital observation (males) and the results were reported using the Tanner scale [23, 24], and grouped according to Finkelstein et al. [25] into early puberty, mid-puberty, and late puberty.

To evaluate skeletal maturation, bone age was obtained by the GP method [26] in which the hand and wrist radiographs were compared to the atlas. This examination was performed under the supervision and guidance of a specialized professional who was unaware of the anthropometric and densitometric characteristics of the adolescents (blind evaluator).

### 2.4 Biochemical evaluation

Blood samples were obtained in the morning after a 10-h fast, between 7:00 am and 9:00 am, and all storage precautions were taken. The samples were collected by well-trained laboratory technicians and evaluated at the Central Laboratory. Dry chemistry was used to assess HDL-c, triglycerides, and fasting glucose in a Vitros 950 analyzer (Johnson and Johnson) according to the manufacturer's instructions. Values are expressed as mg/dL. MetS was defined according to the criteria proposed by the IDF [27]. The adolescents aged 10 to 16 years were classified as having MetS if they presented central obesity, defined by a large waist circumference (>90th percentile of WC by sex and age), and at least two of the four criteria: (TG ≥150 mg/dL, HDL <40 mg/dL, SBP ≥130 mmHg or DBP ≥85 mmHg, and fasting blood glucose ≥100 mg/dL or a type 2 diabetes diagnosis). For both sexes, altered glycemia was the least prevalent MetS factor among the factors analyzed. In the global sample of 271 adolescents evaluated, only 2% of male adolescents and 3% of female adolescents demonstrated glucose intolerance.

The blood samples were collected by venous puncture and centrifuged for 15 min at 1500$g$ for the separation of serum. The serum samples were stored at -70°C until the time of analysis of the biomarkers (BAP, OC, and S-CTx). BAP and intact OC were measured using an assay from Metra™ Biosystems (San Diego, CA, USA), with intra- and inter-assay coefficients of variation of 8 and 7.6%, respectively. S-CTx was quantified by an electrochemiluminescence immunoassay (ECLIA) using the Elecsys beta-Cross Laps serum assay in an automated Elecsys device (Roche™, Indianapolis, IN, USA). The inter-assay coefficient of variation was 5%.

### 2.5 Evaluation of bone mineral density

Bone densitometry was performed in all adolescent participants through dual-energy X-ray absorptiometry (DXA) (Hologic QDR 4500 Discovery A, Hologic, Inc., Bedford, MA, USA). The bone mass results were analyzed using appropriate pediatric software. Bone mineral content is expressed in g and BMD in g/cm$^2$. Measurements were obtained from the L1–L4 lumbar spine region, the total proximal femur including the femoral neck and trochanteric and intertrochanteric regions, and total and subtotal body (whole body less head). Whole body densitometry was performed to obtain total bone mineral content, BMD, and whole body composition. As our densitometry machine has been deactivated, it was impossible to retrieve the results of the BMD for age Z-score, in view of the long time since the evaluations were performed. Thus, we calculated the Lumbar BMD Z-score, Total body BMD, and Subtotal body BMD for age Z-score using the calculator https://zscore.research.chop.edu/bmdCalculator.php [28], to offer a guideline for readers. The effect of body size on bone mass can lead to misinterpretations when comparing individuals of different heights and body compositions. In the current study, the stature and height for age Z-scores did not differ between MetS + and MetS- groups for females (p = 0.167; p = 0.833) and males (p = 0.728; p = 0.881) respectively, and we sought to minimize body size-induced biases in bone mass estimates by using BMD

transformed by body weight (g/cm$^2$/kg body weight) [29]. This transformation can present limitations. Thus, the literature recommends the use of an estimate of volumetric bone mineral apparent density, BMAD (grams per cm$^3$), published by Bachrach et al. [30] and adjusted by Zemel et al. [31]. Zemel et al. [31] use a complex predictive equation of height-for-age Z-score (HAZ) (results not shown). All assessments were performed by only one technician trained for this purpose who had no knowledge of the research. In addition, the instructions of the manufacturer of the device and the standards recommended by the International Society for Clinical Densitometry [32] were followed.

The DXA instrument was calibrated by daily scanning of a hydroxyapatite spine phantom. Machine drift was not observed during the study. The coefficient of variation was estimated from repeated measurements (twice) obtained from 30 patients who are representative of the clinic's patient population for all regions mentioned (lumbar spine and total body) after each patient had been repositioned before scanning. The results demonstrated CVs of 0.6% and 1.3% for the lumbar spine and whole body, respectively. All evaluations were performed by the same blinded, experienced technician, who also performed the densitometry examinations.

## 2.6 Statistical analysis

Data were stored in Excel and analyzed using SAS for Windows v.9.2 software. Normality of the quantitative variables was verified by the Shapiro-Wilk test. For characterization of the sample, descriptive analysis of the data was performed by calculating the distribution of frequencies and measures of central tendency and dispersion. The Student t-test was used for the comparison of chronological and bone age, anthropometric variables (weight, height, BMI), BMD, BMD for age Z-score, transformed BMD, and bone markers between adolescents with and without MetS and its components, stratified by sex. The frequencies of pubertal stages according to sex were compared between adolescents with and without MetS using the chi-square test.

Pearson's correlation was used to evaluate the correlation between serum levels of the markers and variables related to BMD and MetS components. All analyses were performed separately for each sex. Statistical significance was accepted for all analyses when p≤0.05.

## 3. Results

Forty-eight (57%) of the 84 adolescents were female. The mean age was 12.58 ± 1.95 years among girls and 13.42 ± 1.23 years among boys with MetS, both with an average bone age advancement of one year. Among the adolescents, 77.4% were in mid- and late puberty. The adolescents with MetS presented higher weight and BMI than those without MetS (p≤0.05). No significant differences in chronological age, bone age, height, or sexual maturation stage were observed between the matched groups with and without MetS for each sex. The comparison of lumbar spine, proximal femur, and total and subtotal body BMD values, as well as the respective lumbar, total body and subtotal body BMD for age Z-score according to sex and the presence of MetS revealed no significant differences. However, when BMD was transformed to BMD per kilogram of body weight, the female adolescents with MetS exhibited significant decreases in BMD at all sites evaluated (p<0,01) and the males for total and subtotal body BMD (p<0.05) (Table 1).

Regarding the MetS components, male and female adolescents with MetS exhibited a significant increase in waist circumference, serum triglyceride levels, and systolic and diastolic blood pressure. Serum HDL-c levels were lower and significant in the groups with MetS. There were statistically significantly lower concentrations in the three bone biomarkers in female adolescents with excess weight and MetS compared to the matched group. In male adolescents, the mean concentrations of BAP, OC, and S-CTx were also lower in those with MetS, but a significant difference was only observed for BAP and S-CTx (Table 2).

**Table 1. Comparison of adolescents with and without metabolic syndrome according to sex, bone age, tanner stage, anthropometric measures, body mass index, BMD, and transformed BMD of both sexes.**

| Parameter | | Female (n = 48) | | | Male (n = 36) | | |
|---|---|---|---|---|---|---|---|
| | | MetS (-) (n = 24) | MetS (+) (n = 24) | | MetS (-) (n = 18) | MetS (+) (n = 18) | |
| | | Mean ± SD | | p value[a] | Mean ± SD | | p value[a] |
| Age (years) [a] | | 11.78±1.29 | 12.58±1.95 | 0.102 | 13.03±1.30 | 13.42±1.23 | 0.393 |
| Bone age (years)[a] | | 13.09±1.75 | 13.40±2.07 | 0.582 | 13.73±0.92 | 13.86±1.88 | 0.813 |
| *Tanner stage [n (%)][b]* | | *Breast development* | | | *Genital* | | |
| Prepuberty | 1 | 1 (4) | 1 (4.5) | | 0 (0) | 0 (0) | |
| Early Puberty | 2 | 0 (0) | 1 (4.5) | | 2 (14) | 2 (14) | |
| Mid- Puberty | 3 | 8 (35) | 6 (29) | 0.690 | 5 (33) | 4 (29) | 0.994 |
| Late Puberty | 4 | 14 (61) | 13 (62) | | 8 (53) | 8 (57) | |
| | 5 | | | | | | |
| Weight (kg)[a] | | 58.85±9.46 | 73.58±13.92 | <0.000* | 71.32±14.58 | 81.10±13.44 | 0.044* |
| Weight for age Z-score | | 1.53±0.57 | 2.06 ±0.45 | <0.0001* | 1.78 ± 0.64 | 2.19 ± 0.62 | 0.059 |
| Height (m)[a] | | 1.54±0.07 | 1.57±0.07 | 0.167 | 1.63±0.09 | 1.64±0.09 | 0.728 |
| Height for age Z-score | | 0.71±0.88 | 0.66 ±0.86 | 0.833 | 0.72 ± 0.97 | 0.67 ± 1.03 | 0.881 |
| BMI (kg/m$^2$)[a] | | 24.61±2.64 | 29.32±4.16 | 0.038* | 26.44±4.22 | 29.61±4.21 | 0.030* |
| BMI for age Z-score | | 1.54 ± 0.39 | 2.01 ± 0.35 | <0.001* | 1.64 ± 0.49 | 2.00 ± 0.44 | 0.027* |
| Lumbar spine BMD (g/cm$^2$) | | 0.82±0.15 | 0.90±0.19 | 0.123 | 0.77±0.12 | 0.85±0.13 | 0.052 |
| Lumbar BMD for age Z-score | | 0.957 ± 1.35 | 0.874 ± 1.29 | 0.830 | 0.263 ± 1.23 | 0.97 ± 1.08 | 0.073 |
| Femoral BMD (g/cm$^2$) | | 0.89±0.12 | 0.95±0.14 | 0.124 | 0.95±0.16 | 1.00±0.14 | 0.323 |
| Total body BMD (g/cm$^2$) | | 0.90±0.09 | 0.94±0.10 | 0.162 | 0.92±0.08 | 0.94±0.09 | 0.609 |
| Total body BMD for age Z-score | | -0.363 ± 1.492 | -0.06 ± 1.23 | 0.456 | -0.522 ± 1.03 | -0.366 ± 0.94 | 0.641 |
| Subtotal body BMD (g/cm$^2$) | | 0.81±0.08 | 0.85±0.09 | 0.150 | 0.84±0.09 | 0.86±0.09 | 0.560 |
| Subtotal BMD for age Z-score | | -0.259 ± 1.70 | 0.178 ± 1.02 | 0.295 | -0.260 ± 1.11 | -0.154 ± 1.20 | 0.785 |
| Transformed Lumbar spine BMD | | 0.0141±0.0021 | 0.012±0.002 | 0.005* | 0.011±0.002 | 0.010±0.001 | 0.416 |
| Transformed Femoral BMD | | 0.0154±0.0024 | 0.013±0.001 | 0.001* | 0.013±0.002 | 0.012±0.002 | 0.153 |
| Transformed Total body BMD | | 0.0156±0.0023 | 0.013±0.001 | <0.000* | 0.013±0.002 | 0.011±0.002 | 0.042* |
| Transformed Subtotal body BMD | | 0.0141±0.0020 | 0.011±0.001 | <0.000* | 0.012±0.001 | 0.010±0.002 | 0.050* |

MetS (−): without metabolic syndrome; MetS (+): with metabolic syndrome.

BMI, Body Mass Index.

[a] Student t-test.

[b] Chi-square test.

BMD transformed by body weight (g/cm$^2$/kg body weight).

Pearson's correlation coefficients revealed a negative and significant correlation between BMD at all sites and the three bone biomarkers in female adolescents with MetS (Table 3). No significant correlation was observed between BMD and the bone biomarkers in male adolescents (Table 4).

When the same analysis was performed separately for each MetS component among female adolescents with MetS, the increase in waist circumference was negatively correlated with S-CTx, the increase in blood pressure was negatively correlated with BAP and OC, and the

**Table 2. Comparison of metabolic syndrome components and bone biomarkers between adolescents with and without MetS according to sex.**

| Parameter | Female (n = 48) | | | Male (n = 36) | | |
|---|---|---|---|---|---|---|
| | MetS (-) (n = 24) | MetS (+) (n = 24) | | MetS (-) (n = 18) | MetS (+) (n = 18) | |
| | Mean ± SD | | P value[a] | Mean ± SD | | P value[a] |
| *MetS component* | | | | | | |
| WC (cm) | 83.70±7.86 | 94.94±7.48 | < .000 | 86.80±7.35 | 92.68±9.56 | 0.051 |
| HDL-c (mg/dL) | 46.60±10.83 | 39.62±9.41 | 0.022 | 43.77±7.89 | 36.27±3.21 | 0.008 |
| SBP (mmHg) | 104.70±11.06 | 117.30±13.64 | 0.001 | 116.60±9.84 | 128.40±13.70 | 0.005 |
| DBP (mmHg) | 63.77±6.79 | 73.89±8.09 | < .000 | 69.58±5.86 | 80.79±9.00 | 0.000 |
| Triglycerides (mg/dL) | 88.82±19.80 | 149.90±52.03 | < .000 | 99.38±41.74 | 172.60±72.35 | 0.000 |
| Fasting glucose (mg/dL) | 82.78±6.19 | 86.75±8.96 | 0.085 | 87.83±6.90 | 87.22±7.58 | 0.891 |
| *Bone biomarker* | | | | | | |
| BAP (U/L) | 178.10±46.26 | 121.50±65.68 | 0.001 | 244.10 ±56.96 | 179.70±77.46 | 0.008 |
| OC (ng/mL) | 39.08±16.10 | 21.02±12.15 | < .000 | 40.07±12.52 | 31.03±15.59 | 0.066 |
| S-CTx(ng/mL) | 1.56±0.52 | 1.22±0.46 | 0.027 | 1.93±0.33 | 1.60±0.53 | 0.034 |

(−): MetS components without alteration; (+): MetS components with alteration.

WC, waist circumference; HDL-c, HDL-cholesterol; SBP, systolic blood pressure; DBP, diastolic blood pressure; BAP, bone alkaline

Phosphaphosphatase; OC, osteocalcin; S-CTx, carboxy-terminal telopeptide.

[a] Student t-test.

presence of hypertriglyceridemia was positively correlated with BAP and S-CTx. In male adolescents with MetS, the correlation was negative and significant only between increased triglycerides and OC. In adolescents without MetS, a positive and significant correlation was only observed between systolic blood pressure and S-CTx (Table 5).

## 4. Discussion

Comparisons of adolescents with excess weight, with and without MetS, who were similar in terms of chronological age, bone age, and pubertal stage, presented a difference in the concentrations of the bone biomarkers between the two groups. In this respect, adolescents with MetS exhibited significantly lower concentrations of bone biomarkers (BAP, OC, S-CTx) than those

**Table 3. Pearson's correlation between bone biomarkers and bone mineral density measured at four sites in female adolescents with and without metabolic syndrome.**

| Parameter | | MetS (-) | | | MetS (+) | | |
|---|---|---|---|---|---|---|---|
| | | BAP (U/L) | OC (ng/mL) | S-CTx (ng/mL) | BAP (U/L) | OC (ng/mL) | S-CTx (ng/mL) |
| Lumbar Spine | r | -0.04088 | -0.09311 | 0.04937 | -0.71742 | -0.53505 | -0.51012 |
| BMD (g/cm²) | p | 0.873 | 0.665 | 0.818 | < .000 | 0.007 | 0.018 |
| Femoral BMD | r | -0.03186 | -0.07059 | 0.09377 | -0.52271 | -0.68411 | -0.54997 |
| (g/cm²) | p | 0.882 | 0.743 | 0.663 | 0.008 | 0.000 | 0.009 |
| Total body BMD | r | -0.04088 | -0.16939 | 0.10861 | -0.55568 | -0.81760 | -0.53838 |
| (g/cm²) | p | 0.8496 | 0.428 | 0.613 | 0.005 | < .000 | 0.011 |
| Subtotal Body | r | -0.04533 | -0.18300 | 0.12481 | -0.79215 | -0.47089 | -0.54338 |
| BMD (g/cm²) | p | 0.833 | 0.392 | 0.561 | < .000 | 0.023 | 0.010 |

MetS (−): without metabolic syndrome; MetS (+): with metabolic syndrome.

BMD, bone mineral density; BAP, bone alkaline phosphatase; OC, osteocalcin; S-CTx, carboxy-terminal telopeptide.

Pearson correlation test.

**Table 4. Pearson's correlation between bone biomarkers and bone mineral density measured at four sites in male adolescents with and without metabolic syndrome.**

| Parameter | | MetS (-) | | | MetS (+) | | |
|---|---|---|---|---|---|---|---|
| | | BAP (U/L) | OC (ng/mL) | S-CTx (ng/mL) | BAP (U/L) | OC (ng/mL) | S-CTx (ng/mL) |
| Lumbar spine BMD | r | -0.42803 | 0.02406 | -0.07669 | -0.09165 | 0.30822 | 0.01061 |
| (g/cm$^2$) | p | 0.076 | 0.924 | 0.762 | 0.726 | 0.228 | 0.970 |
| Femoral BMD | r | -0.20502 | 0.2714 | 0.22064 | -0.03393 | 0.30356 | -0.03041 |
| (g/cm$^2$) | p | 0.414 | 0.274 | 0.379 | 0.897 | 0.236 | 0.914 |
| Total body BMD | r | -0.31354 | 0.22305 | -0.08190 | -0.07856 | 0.22114 | -0.14209 |
| (g/cm$^2$) | p | 0.205 | 0.373 | 0.746 | 0.764 | 0.393 | 0.613 |
| Subtotal body BMD | r | -0.30173 | 0.22308 | -0.08440 | -0.05501 | 0.15457 | -0.08828 |
| (g/cm$^2$) | p | 0.223 | 0.373 | 0.739 | 0.8339 | 0.5536 | 0.7544 |

MetS (−): without metabolic syndrome; MetS (+): with metabolic syndrome.

BMD, bone mineral density; BAP, bone alkaline phosphatase; OC, osteocalcin; S-CTx, carboxy-terminal telopeptide.

Pearson correlation test.

without MetS, except for OC in male adolescents. Our results showed that adolescents with excess weight and MetS exhibited reduced transformed BMD values at all sites for females, and total and subtotal body densities for males when compared to adolescents without MetS who were matched for chronological age, bone age, and pubertal stage for each sex. In addition, BMD in females was negatively correlated with all bone biomarkers in adolescents with MetS.

Bone mass is acquired gradually during childhood. Acceleration in height and weight growth velocities is observed for females in stages 2–3 of breast and later, and in stages of pubertal development 3–4 for males, with the peak bone mineral accretion rate occurring on average seven months or slightly later than PHV in both sexes [23, 24, 33]. According to the National Osteoporosis Foundation, the age group in which peak velocity of bone acquisition is achieved is approximately 12.5 ± 0.90 years for female adolescents and 14.1 ± 0.95 years for male adolescents [23, 24, 33]. Thus, the adolescents in our study are probably at the opportune age for maximum peak velocity of bone acquisition, especially the female adolescents.

We have previously reported that adolescents with excessive weight and MetS exhibited reduced transformed BMD [5, 29] values at all sites (lumbar spine (L1-L4), proximal left femur, and total and subtotal) compared to adolescents without MetS [5] and the results of the current study corroborate those previously published. In a previous study [5] female adolescents with a large waist circumference, low HDL-c, hypertriglyceridemia, and high blood pressure showed significant reductions in BMD at all sites evaluated (p<0.01) and this pattern was also observed in male adolescents, with the exception of the increase in triglycerides (p>0.05). These results suggest that each MetS component might act independently on BMD in a sex-specific manner [5]. Confirming our evidence, a recent study evaluated 306 female adolescents aged 9 to 12 years, analyzing the isolated and combined effect of cardiometabolic risk factors (CMR) (calculated by the CMR z-score of the included sample, where CRM z-scores were created from the logarithmic transformed form of the variables: homeostatic model assessment of insulin resistance (HOMA-IR), C-reactive protein (CPR), triglycerides (TG), high-density lipoprotein cholesterol (HDL-c), low-density lipoprotein cholesterol (LDL-c), and mean arterial blood pressure (MAP)) on bone mass using DXA together with the evaluation of bone geometry, vBMD, and bone strength through the use of peripheral quantitative computed tomography (pQCT) in the tibia and femur. The authors identified that the CMR z-score presented significant inverse associations with DXA total body BMC and bone area and with

**Table 5. Pearson's correlation between bone biomarkers and metabolic syndrome components in adolescents of both sexes with and without MetS.**

| MetS component | | MetS (-) | | | | MetS (+) | | |
|---|---|---|---|---|---|---|---|---|
| | | BAP (U/L) | OC (ng/mL) | S-CTx (ng/mL) | | BAP (U/L) | OC (ng/mL) | S-CTx (ng/mL) |
| | | *Female* | | | | | | |
| WC (cm) | r | -0.29836 | -0.07739 | -0.24094 | | -0.32473 | -0.24808 | -0.46752 |
| | p | 0.177 | 0.732 | 0.280 | | 0.121 | 0.242 | 0.032* |
| HDL-c (mg/dL) | r | 0.13497 | -0.0581 | -0.11359 | | -0.01574 | -0.28168 | -0.00502 |
| | p | 0.539 | 0.792 | 0.605 | | 0.941 | 0.182 | 0.982 |
| SBP (mmHg) | r | -0.22598 | -0.0187 | -0.16776 | | -0.50237 | -0.46993 | -0.17346 |
| | p | 0.311 | 0.934 | 0.455 | | 0.014* | 0.023* | 0.464 |
| DBP (mmHg) | r | -0.21424 | 0.00212 | -0.20241 | | -0.35271 | -0.4547 | -0.02847 |
| | p | 0.338 | 0.992 | 0.366 | | 0.098 | 0.029* | 0.905 |
| Triglycerides (mg/dL) | r | -0.40542 | -0.12222 | -0.19898 | | 0.46519 | 0.20343 | 0.51017 |
| | p | 0.055 | 0.578 | 0.362 | | 0.022* | 0.340 | 0.018* |
| Fasting glucose (mg/dL) | r | -0.00579 | -0.22448 | -0.23428 | | -0.09644 | -0.0954 | -0.17617 |
| | p | 0.979 | 0.303 | 0.281 | | 0.654 | 0.657 | 0.444 |
| | | *Male* | | | | | | |
| WC(cm) | r | 0.45455 | 0.19324 | 0.11532 | | 0.14272 | 0.39229 | 0.19134 |
| | p | 0.058 | 0.442 | 0.648 | | 0.598 | 0.132 | 0.512 |
| HDL-c (mg/dL) | r | 0.00244 | 0.29945 | 0.05388 | | -0.33351 | -0.18099 | -0.07285 |
| | p | 0.992 | 0.227 | 0.831 | | 0.190 | 0.487 | 0.796 |
| SBP (mmHg) | r | 0.02191 | 0.30439 | 0.34550 | | 0.15801 | 0.20069 | 0.37223 |
| | p | 0.931 | 0.219 | 0.160 | | 0.558 | 0.456 | 0.190 |
| DBP (mmHg) | r | 0.30301 | 0.05582 | 0.53446 | | 0.24027 | 0.14589 | 0.26202 |
| | p | 0.221 | 0.825 | 0.022* | | 0.370 | 0.589 | 0.365 |
| Triglycerides (mg/dL) | r | -0.20649 | -0.11482 | 0.22251 | | -0.22418 | -0.48347 | -0.15625 |
| | p | 0.411 | 0.650 | 0.374 | | 0.387 | 0.049* | 0.578 |
| Fasting glucose (mg/dL) | r | -0.19083 | -0.1420 | 0.05144 | | 0.17677 | 0.13868 | -0.14624 |
| | p | 0.448 | 0.568 | 0.839 | | 0.497 | 0.595 | 0.603 |

MetS (−): without metabolic syndrome; MetS (+): with metabolic syndrome.

BAP, bone alkaline phosphatase; OC, osteocalcin; S-CTx, carboxy-terminal telopeptide; WC, waist circumference; HDL-c, HDL-cholesterol; SBP, systolic blood pressure; DBP, diastolic blood pressure. Pearson correlation test.

pQCT regional bone measures with total and trabecular bone area, highlighting that the set of CMR factors had a negative effect on bone mass, not evident for all components individually [6].

Researchers have highlighted the importance of understanding the evolution of BMD and alterations in the concentrations of bone biomarkers during adolescence [34, 35]. With respect to indicators of bone metabolism, a sample with 101 eutrophic female adolescents was evaluated, and the results demonstrated higher mean biomarker concentrations when the adolescents were under the age of 16, while concentrations of bone biomarkers were lower at the end of adolescence (after 16 years) [34]. Similarly, in male adolescents, the serum biomarker concentrations differed significantly among age groups, with the highest concentrations of the biomarkers before 16 years of age and the lowest concentrations after 16 years of age [35]. Similar results were recently published with the follow-up of 2,416 children and adolescents, analyzing the bone markers osteocalcin, PINP, and CTX-I. In total, 317 of the participants were obese and the remainder eutrophic [36]. During puberty, the highest concentrations of bone markers occurred from 10 to 11 years in girls, and 13 years for boys. These results are very close to

those presented by Fortes et al. [34] and Silva et al. [35] in terms of age, revealing a reduction in biomarkers after age 16 in both sexes [36].

Within this context, proportionally higher levels of bone biomarkers would be expected in the adolescents in the is present study, since the mean CA of girls with or without MetS was around 12 years of age and BA, around 13 years, while in boys both CA and BA were around 13 years of age. The concentrations obtained among adolescents without MetS in the present study were close to those observed among the eutrophic individuals analyzed in the studies cited [34, 35]. However, significant reductions in both biomarkers of bone formation and resorption were observed in the group of adolescents with MetS compared to the matched group without MetS.

Studies in adolescents evaluating the impact of MetS on biomarkers of bone formation and resorption are sparse. In recent years, concern has been directed towards the biomarker OC. In the adult population, researchers found that the accumulation of visceral fat, lipid profile alterations, and altered blood pressure, the main components involved in MetS, were correlated with low serum levels of OC [37]. With respect to the relationship between MetS and OC, Tan et al. [38] evaluated 2,344 Chinese men between 20 and 69 years of age and found a negative association between OC levels and the presence of MetS. These results agree with our findings, however, in our results the reductions in OC were significant and negative only in female adolescents with MetS. Thus, analyzing the concentrations of bone biomarkers and the evolution of BMDs according to the sexes in the present study (Table 2), there was a predominance of significant differences for the female sex, with the presence and absence of MetS, which may be related to the nutritional evaluation and biochemical alterations observed between them. In the group of those without the presence of Metabolic Syndrome, 42% of the patients presented no biochemical alterations (HDL-c, triglycerides, and fasting glucose) and weight (kg) and BMI ($kg/m^2$) (46%) between ≥85th percentile and <95th percentile when compared to those with the presence of MetS. On the other hand, in the group of adolescent males, even among those without MetS, it was observed that 83% presented biochemical alterations (HDL-c, triglycerides, and fasting glucose) and weight (kg) and BMI ($kg/m^2$) higher or equal to the 95th percentile were observed in 55.5% of them. This finding leads to the speculation that the lack of detection of significant differences for the set of biomarkers among male adolescents, when the MetS and absence of MetS groups were compared, was due to the fact that although they were not classified with the presence of MetS, they were much more compromised metabolically and nutritionally than the female adolescents with the absence of MetS.

The literature states that osteocalcin, besides being a bone-derived hormone, is also associated with energy metabolism. This association is negative (inverse) against hyperglycemia, insulin resistance, hypertriglyceridemia, HDL-c reduction, etc. [38], confirming the connection between bone and energy metabolism [39]. In addition, BAP is also associated with MetS [40], with strong positive associations with hyperinsulinemia and insulin resistance and negative and significant associations with other MetS components such as HDL-c, tending to significance with waist circumference and triglycerides, showing distinct associations with multiple components of MetS [40].

Considering the high prevalence of osteoporosis, obesity, and MetS in adults, Laurent et al. [41] evaluated the relationship of MetS with bone turnover in 3,129 men between 40 and 79 years of age. MetS was present in 975 men (31.2%). The men with MetS demonstrated significantly lower levels of all bone biomarkers and higher BMD at the hip, femoral neck, and lumbar spine. When the MetS components were analyzed individually, only hypertriglyceridemia and hyperglycemia were inversely associated with N-terminal propeptide of type I procollagen (PINP) and S-CTx. Osteocalcin was independently and inversely associated only with

hyperglycemia. Men with three, four, or five MetS criteria had lower bone turnover markers. On the other hand, men with one or no criteria presented higher bone turnover markers, indicating the absence of a limit of three criteria above which these markers would be altered. The authors concluded that MetS is associated with lower bone renewal in adult men.

In agreement with the results of the aforementioned studies, in a population of adolescents, it was demonstrated that MetS alters bone remodeling [3–5] but not all MetS components show similar associations.

Evidence suggests that MetS favors the differentiation of adipocytes and suppresses the differentiation of osteoblasts, reducing bone formation activity. Furthermore, chronic inflammation induced by MetS could increase the formation of osteoclasts and consequently their bone resorption activity [42, 43]. However, in the present study there was a reduction in the concentrations of all bone markers analyzed in both sexes among adolescents with MetS, suggesting an unfavorable effect on bone formation and resorption, possibly indicating a reduction in remodeling. We emphasize that in normal situations the remodeling starts through the resorption that precedes the bone formation, thus in these adolescents with MetS both processes are affected, indicating a low turnover rate.

With respect to MetS and bone remodeling, studies have reported a chronic inflammatory response caused by inflammatory cytokines produced by visceral adipose tissue in obese individuals, which was associated with the process of osteoclastogenesis when bone resorption would be higher. This chronic inflammatory response possibly contributes to the reduction in bone mass acquisition during puberty in obese adolescents [44].

Added to this, a recent study indicated negative associations between leptin and OC, suggesting that the inflammatory state may negatively influence bone turnover markers in obese adolescents [45, 46].

There is a limited number of prospective trials, either cohort or clinical, and descriptive studies investigating the evolution of bone biomarkers in adolescents with excess weight and with a diagnosis of MetS. To our knowledge, this is the first study demonstrating the impact of MetS on bone remodeling in adolescents. Thus, in view of the scarce results in the literature, the combined analysis of biochemical markers of bone formation and resorption in a matched sample selected using rigorous inclusion criteria can be considered a strength of the present study. Regarding the cross-sectional design of the study, this prevents the provision of information about the temporal sequence of events and does not permit identification of causal relationships. Previous studies demonstrated, through densitometry, a reduction in bone mass between those who are obese and severe/extremely obese and those with the presence of MetS [5, 47], thus we propose that bone turnover markers reveal a reduction in bone remodeling in adolescents with MetS. It is suggested that both procedures are useful and complementary in bone assessment in adolescents.

The current study has some limitations, among which we did not evaluate the pro and anti-inflammatory cytokines of the adolescents, which are possibly involved in MetS. In addition, serum levels of 25-hydroxyvitamin D (s25 (OH) D) were not measured, and a deficiency/insufficiency in this biomarker has been described in association with the presence of MetS components in children [48], adolescents, and adults [49]. However, this study contributed by suggesting possible designs for future studies, especially prospective studies, with the inclusion of a larger number of subjects and the possibility of generalizing the results.

In summary, the inclusion and interpretation of this set of bone biomarkers and of others to be incorporated in the future, will complement and broaden knowledge about bone metabolism in adolescents with and without MetS.

## 5. Conclusion

Our results suggest that adolescents with excess weight and the presence of MetS presented lower concentrations of biomarkers of bone formation and resorption. The presence of MetS could be unfavorable to the development of bone mass in adolescents. The reduction in BMD and in the markers responsible for bone formation during this phase of life may compromise the acquisition of peak bone mass, which could increase the current fracture risk and result in bone microarchitecture derangement and bone fragility in the years to come.

## Supporting information

**S1 Data.**
(XLSX)

## Author Contributions

**Conceptualization:** Tamara Beres Lederer Goldberg, Carla Cristiane Silva, Cilmery Suemi Kurokawa, Luciana Nunes Mosca Fiorelli, Anapaula da Conceição Bisi Rizzo.

**Data curation:** Luciana Nunes Mosca Fiorelli, Anapaula da Conceição Bisi Rizzo.

**Formal analysis:** José Eduardo Corrente.

**Funding acquisition:** Tamara Beres Lederer Goldberg.

**Investigation:** Valéria Nóbrega da Silva, Tamara Beres Lederer Goldberg, Luciana Nunes Mosca Fiorelli, Anapaula da Conceição Bisi Rizzo.

**Methodology:** Valéria Nóbrega da Silva, Tamara Beres Lederer Goldberg, Anapaula da Conceição Bisi Rizzo.

**Project administration:** Valéria Nóbrega da Silva, Luciana Nunes Mosca Fiorelli.

**Resources:** Tamara Beres Lederer Goldberg.

**Software:** José Eduardo Corrente.

**Supervision:** Tamara Beres Lederer Goldberg, Cilmery Suemi Kurokawa, José Eduardo Corrente.

**Validation:** Carla Cristiane Silva, Cilmery Suemi Kurokawa.

**Writing – original draft:** Valéria Nóbrega da Silva, Tamara Beres Lederer Goldberg, Carla Cristiane Silva, Cilmery Suemi Kurokawa.

**Writing – review & editing:** Valéria Nóbrega da Silva, Tamara Beres Lederer Goldberg, Carla Cristiane Silva, Cilmery Suemi Kurokawa, José Eduardo Corrente.

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
