## [Decision Letter · Decision Letter 0]

23 Nov 2020

PONE-D-20-24612

Impact of metabolic syndrome and its components on bone remodeling in adolescents

PLOS ONE

Dear Dr. Goldberg,

Thank you for submitting your manuscript to PLOS ONE. After careful consideration, we feel that it has merit but does not fully meet PLOS ONE’s publication criteria as it currently stands. Therefore, we invite you to submit a revised version of the manuscript that addresses the points raised during the review process.

Editor’s Comments 

**Title:** Impact of metabolic syndrome and its components on bone remodeling in adolescents

Goldberg et al have submitted a manuscript comparing BMD and bone turnover markers in overweight/obese teenagers with metabolic syndrome vs those without metabolic syndrome. The concluded that ‘adolescents with MetS exhibited significantly lower concentrations of bone biomarkers (BAP, OC, and S-CTX) than those without MetS, except for OC in male adolescents. In addition, BMD was negatively correlated with all bone biomarkers in adolescents with MetS’.

This is an interesting area of exploration in pediatric bone metabolism.

**Comments:**

The manuscript is quite voluminous and some of the sections such as the Introduction and Discussion should be edited and shortened considerably for clarity.Authors should attempt to offer an explanation why all the bone turnover markers were decreased in the cohort with MetS.The authors should also endeavor to explain the reason for the negative correlation of all bone markers (both bone formation and bone resorption markers) with BMDThe finding of reduced bone formation markers and BMD in MetS subjects suggests that the underlying mechanism or the reduced biomarkers and BMD is most likely due to increased systemic inflammation which was not explored in this study. To establish this, however, will require a study of the patients’ pro-inflammatory and anti-inflammatory cytokines which was not done in this study. Therefore, this should be discussed in the section on limitations of the study.It was not clear why the authors did not evaluate vitamin D status in their subjects which could have shed some light on their findings. This is a significant limitation of the study and should be discussed in the section on limitations.Table 1 should be completely redone. Table 1should present the comparison between Mets patients and non-Mets patients for anthropometry and biochemical parametersPlease review the comments from the reviewers below and provide the anthropometric and radiological data in z-scores where appropriate.

We look forward to receiving your revised manuscript.

Kind regards,

Benjamin Udoka Nwosu, MD

Academic Editor

PLOS ONE

Journal Requirements:

2.  Please provide additional details regarding participant consent. In the ethics statement in the Methods and online submission information, please ensure that you have specified what type of consent you obtained (for instance, written or verbal, and if verbal, how it was documented and witnessed).”

3. In your Methods section, please provide additional information about the participant recruitment method and the demographic details of your participants. Please ensure you have provided sufficient details to replicate the analyses such as: a) the recruitment date range (month and year) and b) a description of how participants were recruited.

4. Please include your tables as part of your main manuscript and remove the individual files. Please note that supplementary tables (should remain/ be uploaded) as separate "supporting information" files.

Reviewers' comments:

Reviewer's Responses to Questions

**Comments to the Author**

1. Is the manuscript technically sound, and do the data support the conclusions?

Reviewer #1: Partly

Reviewer #2: Partly

2. Has the statistical analysis been performed appropriately and rigorously? 

Reviewer #1: Yes

Reviewer #2: Yes

3. Have the authors made all data underlying the findings in their manuscript fully available?

Reviewer #1: No

Reviewer #2: No

4. Is the manuscript presented in an intelligible fashion and written in standard English?

Reviewer #1: No

Reviewer #2: Yes

5. Review Comments to the Author

Reviewer #1: Comment 1: The study had a decent cohort size of adolescents matched by age, bone age, and puberty. In addition, it was stratified by sex. The authors compared adolescents with MetS with adolescents without MetS as controls. It would also be interesting how these findings relate to normal weight adolescents.

Though evaluation of obesity and MetS and BMD in pediatrics has been published, the authors took this further to attempt to look at affects of MetS in adolescents with excess weight on bone remodeling markers, which is a very interesting and important concepts.

The conclusion, objective in introduction worded strongly. BMD and bone mass used interchangeably in the conclusion. BMD data and analyses not provided between the groups. BMD data was provided in tables showing r correlations with bone resorption markers only. Can the authors show BMD data and analysis done to draw the main discussion and conclusion points?

Was DXA measurements adjusted for height? please explain why or why not.

Authors report BMI, but no BMI %iles or SDS. In pediatric data report the latter to categorize the cohort.

In the lines 296-298, the authors state that the study showed a significant reduction in BMD at different sites. It is not clear what the data were. The authors should clarify and show BMD data to support the statement.

Lines 312-319:The authors have an interesting finding pertaining to the sex differences,

they speculate that it may be attributable to the differences in biochemical and anthropometric characteristics. It is not clear what biochemical characteristics are being referred to.

Similarly, line 322-323, states males were much more compromised metabolically, however,

it is not clear in which sense and what metabolic derangements this assessment is being referred to.

In the conclusion session, authors mention the reduction in BMD and its effects on peak bone mass, however, there are no data presented on BMD values or Z-scores in this cohort.

MetS increases bone resorption, however, CTX was lower in MetS group indicating lower bone resorption. It is eluded that most likely due to decrease bone remodeling. However in chronic inflammation states of MetS and obesity osteoclast and bone resorption activities are significantly elevated as been shown in other adults studies and sites by the authors. In addition, authors showed that BMD negatively correlated with CTX. Can authors elaborate further on these rather unexpected findings?

Table 4 appears to have a small shift in the values, which should be an easy edit.

Table 5 is difficult to read, p values are missing in the comparison of Lumbar spine BMD with the bone makers; instead, r values are provided twice.

Table 6 is difficult to read and follow, recommend revision. There are multiple comparisons in the study, especially in table 6. It is not clear if correction for multiple comparisons was done. Can authors comment?

Comment 3: Besides tables and results in the manuscript no additional data were provided. BMD data were not provided.

Comment 4: Discussion had comprehensive overview and review of available data on this and relevant topics.

Many sentences are too long and difficult to follow. Discussion should be revised and edited for clarity.

Line 59: "non-definitively demonstrated" - consider revision

Sentence in lines 261-268 is not clear and seems ambiguous. please revise the point.

In line 268 authors use BMTs, I am assuming it's a typo and they mean BMD. This should be clarified and corrected. There was another area where BMTs were used in place of intended BMD.

Lines 303-305: the sentence is incomplete. this needs a grammar check.

The statements in lines 332-338 need references.

The sentence in lines 417-424 is confusing and very long, needs revision.

Reviewer #2: General Comments

This appears to be a sub-study of a larger study looking at the impact of metabolic syndrome and its components on bone remodeling in adolescents. The authors present relevant data and the use of biochemical markers of bone turnover is appreciated given the age group and lack of research using bone markers in adolescents particularly in those who are classified as overweight or obese. This data would be a welcome addition to the literature.

There are number of general and specific comments that need to be addressed with respect to the interpretation of the bone remodeling data, editing of wording, and the discussion as it is currently written. For these reasons, a major revision is recommended.

As well, the Authors have a paper published in Bone (2014) examining MetS and its effects on bone mass, similarly using 271 participants and study design. Although that paper did not measure bone markers, it is not included in the current reference list even though sections of the methods could refer to it and make it clear if the 271 participants are similar between both papers. It appears the present study is a subset or additional analysis of a larger one that may have already been published. Are the current 84 participants from the original 271 participants or have any of them been additionally recruited? Can the authors provide any information or discussion on the use of or lack thereof a control group (normal weight non-MetS).

Much of the discussion is spent on studies and finding relating to OC, however that is not the only bone marker that was measured in the current study. CTx was mentioned briefly and there was no discussion provided to BAP. In addition, the authors provide meaningful information on types of diet, inflammation and cytokines in the discussion, but fail to use this information to support and make connections to the results of their study. Given that inflammatory cytokines or RANKL were not measured in the current study, I would recommend the authors not include these sections, unless they can connect it with the variables they did in fact measure. I would suggest speaking more to the specific MetS components, the number of them present and addressing S-CTx and BAP in the discussion.

Specific Comments

Abstract

1. Numerical values or results in terms of r- or p-values should be presented in the abstract.

Introduction

1. Define MetS, the conditions it can include, risk factors and how this relates to and presents in adolescents. How many MetS components do adolescents need to be classified as having MetS?

2. In the second paragraph of the introduction (lines 62-69), make reference to bone development papers. Only one reference is used at the end of the paragraph.

3. Line 66, the use of the word “intense” to describe the imbalance between bone formation vs resorption due to growth should be rephrased.

4. Line 67-68, “under normal physiological conditions” and “dependent phenomena”, is this in reference to growth in adolescents or a healthy adult? How bone remodeling is occurring during adolescence as a result of growth and maturation needs to be considered when describing this process.

5. Line 74 mentions a systematic review, please provide reference.

6. Line 85, what is meant by “specialized literature”?

7. It is mentioned, line 80, that components of MetS are negatively correlated with BMD in adolescents. It may be worth including and referencing literature that has demonstrate negative impacts of excess weight in adolescents on BMD, BMC and/or bone strength, in addition to the incidence of increased fracture. There is some literature that has demonstrated negative correlations between BMI or excess weight on marks of bone turnover. This can and should be used to support some of the findings related to MetS and bone properties.

Materials and Methods

1. Please refer to general comment as it pertains to previously published paper that could be referenced in this section of the document.

2. Final paragraph of section 2.1 (lines 132-136) describe how participants were matched and compare to the 271 subjects. Consider moving this to the first paragraph of section 2.1 to make it clear earlier in the paper how participants were matched for the study analysis.

3. Why did the authors choose to match participants by chronological age, bone age, and pubertal stage? Please provide reference or justification for using these matching criteria and perhaps for not including anthropometric measures.

4. Are the authors aware and able to report how many of the female participants were pre vs. post-menarcheal, or had all female participants reached menarche?

5. It is not clear how the information that was used for the inclusion criteria was collected. Was this done by interview, survey, or a specific questionnaire. Specifically, how was the nutritional information collected?

6. Line 141, potential editing error in sentence – “utilized to classify nutritional status”.

7. Like 143 makes refence to growth curves. Are these based on WHO or CDC curves?

Statistical Analysis/Results

1. Given that DXA was used, is it possible for the authors to provide data on lean and fat mass, or as a percentage, for the participants. Is there BMC data that can be reported?

2. Table 2, the n-values for the Tanner stages do not add up to be the n-values for the respective group.

3. It is not clear what Tanner stages the authors are using to define participants as early, mid or late puberty. Please define this in the maturation section of the methods.

4. Statistical analysis section states (Line 215-218) than an ANOVA was conducted to compare bone markers with the number of MetS components, however results are not presented in the document.

5. In terms of analysis, the age range (10-16yrs) in addition to the inclusion of all tanner stages into the analysis may be problematic and contributing to some of the correlations and findings.

A) Bone marker concentrations change with maturity and there have been studies to show varying levels by tanner stage. I would expect to see negative correlations since BMD increases and bone marker concentrations decrease with age/maturation. It is not clear from the data and subsequent discussion why this relationship is significant in females and more so in those that have MetS. Is the number of MetS components, or a specific component, or perhaps their excess weight (percent body fat) a factor? Is it related to their maturation? Looking at the data more females are in late puberty compared to boys. Is the menarcheal status between the female MetS and non-MetS participants similar?

B) Because of the changing bone marker concentrations with maturation it is not uncommon for studies to separate participants into groups based on tanner staging. Have the authors considered creating 2 tanner groups to help minimize maturation effects that come with having all tanners 1-5 grouped together?

C) As well, given that BMD is affected by body size, have the authors considering or tried adjusting their analysis by controlling for various anthropometric measures (i.e. height)?

Discussion

1. Line 256 – in reference to the “behavior”’ of bone biomarkers. Please rephrase to be clearer as to what the results were.

2. Line 259 – states that BMD was negatively correlated with all bone markers in adolescents with MetS. However, this was only the case with females and not males. Please correct this statement.

3. Lines 261-268 – authors discuss collinearity of the bone markers. What is the purpose of this finding and how does it relate to the literature and data?

4. Line 268 – what is BMT?

5. Provide reference at end of sentence (line 271)

6. Paragraph (Line 277-295) – Authors discuss bone marker levels by age and tanner stage. Please be specific if you are referring to males vs. females in those studies. And which of these studies used participants with MetS. As well, how does these studies and their grouping compare given that the current study includes all tanner stages into one group. Please refer to comment 5b in previous section.

7. Line 307 – Unclear what the results of Tan et al were and how they compare to the current findings. Please elaborate.

8. Line 318 – what do the authors mean by “biochemical alterations”?

9. Paragraph (lines 324-327) – small paragraph talking about OC. Consider reorganizing and placing it within previous paragraph where authors discuss OC. As well, please provide clearer discussion of findings when comparing current study results with study using eutrophic children, when such a group was not used in the current study.

10. Lines 334-335 – please provide references

11. It is not clear the connection the authors are trying to make between different types of OC and various other physiological markers, and how this relates to the finding of the current study.

12. Please provide discussion on limitations for the current study. Please include some methodological limitations of using DXA measurement with children and adolescents.

13. Line 426 and 427 – the authors mention “raising new hypotheses” and suggesting possible “design for future studies”, although hypotheses were not presented within the document. Can the authors please include some examples for potential future studies or considerations.

6. PLOS authors have the option to publish the peer review history of their article (what does this mean?). If published, this will include your full peer review and any attached files.

Reviewer #1: No

Reviewer #2: No

---

## [Author Response · Author response to Decision Letter 0]

8 Jan 2021

To Benjamin Udoka Nwosu, MD

Academic Editor

PLOS ONE

Dear Dr. Udoka Nwosu

First, we are grateful that you considered the topic of this manuscript to be an interesting area of exploration in pediatric bone metabolism and for sending your personal comments and the excellent comments from the reviewers regarding our article "Impact of metabolic syndrome and its components on bone remodeling in adolescents". We have carefully read and reread the comments to ensure that we have effectively responded to the suggestions and questions raised. Below are the comments made by the reviewers and, in sequence, our answers, highlighted in blue. The passages added to the original article are also highlighted in blue.

We would like to take this opportunity to thank you for the possibility of re-presenting the study to PLOS One with the incorporation of the comments made by the reviewers.

Comments:

1. The manuscript is quite voluminous and some of the sections such as the Introduction and Discussion should be edited and shortened considerably for clarity. 

Answer: We have reduced the two sections as suggested.

2. Authors should attempt to offer an explanation why all the bone turnover markers were decreased in the cohort with MetS.

Answer: An explanation has been added to the discussion.

3. The authors should also endeavor to explain the reason for the negative correlation of all bone markers (both bone formation and bone resorption markers) with BMD.

Answer: Dear Editor, in previous studies from our group with eutrophic adolescents, we observed negative correlations between bone markers and bone mineral density in females. We have written:

“Bone remodeling biomarkers (BAP, OC, and S-CTx) presented a significant negative correlation with CA, BA, and pubertal stage. This finding demonstrated that the more mature the participants, the higher their BMD values. Through analyses of biomarkers, that study demonstrates the changes in bone remodeling occurring in the second decade of life, revealing high marker concentrations in the early adolescence years and significantly reduced concentrations in late adolescence. These analyses correlate with the BMD values, which represent bone mass incorporation, and indicate an inversely proportional behavior showing the highest BMD values associated with the lowest concentrations of formation and resorption of biomarkers (Fortes et al., 2014).”

Thus, this negative correlation represents the expected evolution in eutrophic adolescents and, possibly, in those with MetS. However, we highlight that the concentrations of the markers were even lower in the group with MetS and there were stronger correlations between biomarkers and bone mineral density, as shown in the manuscript. Although the increase in BMD was less than expected (Nóbrega da Silva et al., 2014), a negative correlation was found only in girls with bone markers. Similar results were found in a previous study (Yilmaz et al., 2005) which reported: “A negative correlation of BMD at two sites with bone formation markers was significant in girls (p< 0.001), but no correlation could be detected in boys (p> 0.05)”.

References:

Fortes CM, Goldberg TB, Kurokawa CS, Silva CC, Moretto MR, Biason TP, Teixeira AS, Nunes HR. Relationship between chronological and bone ages and pubertal stage of breasts with bone biomarkers and bone mineral density in adolescents. J Pediatr (Rio J). 2014;90(6):624-31. 

Nóbrega da Silva V, Goldberg TBL, Mosca LN, Rizzo ACB, Teixeira AS, Corrente JE. Metabolic syndrome reduces bone mineral density in overweight adolescents. Bone. 2014; 66:1-7.

Yilmaz, D., Ersoy, B., Bilgin, E. et al. Bone mineral density in girls and boys at different pubertal stages: relation with gonadal steroids, bone formation markers, and growth parameters. J Bone Miner Metab.2005; 23:476–482.

4. The finding of reduced bone formation markers and BMD in MetS subjects suggests that the underlying mechanism or the reduced biomarkers and BMD is most likely due to increased systemic inflammation which was not explored in this study. To establish this, however, will require a study of the patients’ pro-inflammatory and anti-inflammatory cytokines which was not done in this study. Therefore, this should be discussed in the section on limitations of the study. 

Answer: You are correct. We have removed the text that introduced this possibility about the inflammatory state and presented it as one of the limitations of the present study.

5. It was not clear why the authors did not evaluate vitamin D status in their subjects which could have shed some light on their findings. This is a significant limitation of the study and should be discussed in the section on limitations.

Answer: We agree that this is another limiting factor in the study because 25-hydroxyvitamin D [25 (OH) D] was not dosed. However, as all the adolescents included and evaluated were overweight and the groups were paired, we believe that both groups had Vitamin D insufficiency/deficiency, whether or not they had MetS.

6. Table 1 should be completely redone. Table 1should present the comparison between Mets patients and non-Mets patients for anthropometry and biochemical parameters.

Answer: Dear Editor, Table 1 corresponded to the IDF criteria for the classification of Metabolic Syndrome and has been removed, and the description of the components of Metabolic Syndrome has been added to the manuscript. 

The current table 1 is entitled : Comparison of adolescents with and without metabolic syndrome according to sex, bone age, tanner stage, anthropometric measures, and body mass index, and BMD and transformed BMD of both sexes have been added . 

The bone markers are included in table 2, entitled: Comparison of metabolic syndrome components and bone biomarkers between adolescents with and without MetS according to sex.

7. Please review the comments from the reviewers below and provide the anthropometric and radiological data in z-scores where appropriate.

Answer: Dear Editor, each Reviewer's comment has been carefully analyzed and we have provided detailed answers below.

Review Comments to the Author

1.Reviewer #1: Comment 1: The study had a decent cohort size of adolescents matched by age, bone age, and puberty. In addition, it was stratified by sex. The authors compared adolescents with MetS with adolescents without MetS as controls. It would also be interesting how these findings relate to normal weight adolescents. Though evaluation of obesity and MetS and BMD in pediatrics has been published, the authors took this further to attempt to look at affects of MetS in adolescents with excess weight on bone remodeling markers, which is a very interesting and important concepts.

Answer: Dear Reviewer, we appreciate your encouraging comments and your careful analysis of our study. We have tried to answer each question in order to improve the manuscript. 

2. The conclusion, objective in introduction worded strongly. BMD and bone mass used interchangeably in the conclusion. BMD data and analyses not provided between the groups.

Answer: You are absolutely right, we included a table without the reported BMD results evaluated in the various analyzed sites. We have changed table 1 and included the BMD means, as well as the transformed BMDs. 

The conclusion has been changed to: The presence of MetS is detrimental to the bone mineral density (BMD) and biochemical markers of bone formation and resorption in adolescents with excess weight.

3. BMD data was provided in tables showing r correlations with bone resorption markers only. Can the authors show BMD data and analysis done to draw the main discussion and conclusion points?

Answer: We have added the means and SD of the BMD to Table 1. Tables 3 and 4 show BMD and Biomarker values (OC, BAP, and S-CTx) for boys and girls respectively.

4.Was DXA measurements adjusted for height? please explain why or why not.

Answer: Densitometry was not adjusted for height, but for weight. This is the same technique that we used in a manuscript that we previously published in BONE (Nóbrega da Silva et al., 2014) citing the study of Hill et al. (2011). As the groups were matched for CA, BA, and pubertal development, they did not differ in height, as can be seen in Table 1.

References:

Nóbrega da Silva V, Goldberg TB, Mosca LN, Bisi Rizzo Ada C, Teixeira Ados S, Corrente JE. Metabolic syndrome reduces bone mineral density in overweight adolescents. Bone. 2014;66:1-7. 

Hill KM, Braun MM, Egan KA, Martin BR, McCabe LD, Peacock M, et al. Obesity augments calcium-induced increases in skeletal calcium retention in adolescents. J Clin Endocrinol Metab 2011 ;96:2171–7.

5. Authors report BMI, but no BMI %iles or SDS. In pediatric data report the latter to categorize the cohort.

Answer: You are absolutely right about the percentile assessment. This analysis was performed for each of the individuals included in the study, as we explain in the methods. However, due to the number of words, we did not include it in the text or in a table, as we understood that we would have a lot of data and that this was not the focus of the study, since we only included overweight individuals. We present the nutritional classification according to percentiles of the groups paired by CA, BA with MeTs - and MeTS + only for information, remembering that adolescents with a BMI between the 85th and 95th percentile were classified as overweight, between the 95th and 99th percentile as obese (Kuczmarski, Ogden, Guo, 2002), and above the 99th percentile as extremely obese (Barlow, 2007).

Nutritional evaluation of adolescents matched and included in the study sample.

 Female (n=48) Male (n=36)

 MetS (-) MetS (+) MetS (-) MetS (+)

Overweight 11 (45.9%) 4 (16.7%) 8 (44.4%) 3 (16.7%)

Obese 11 (45.9%) 14 (58.3%) 8 (44.4%) 5 (27.8%)

Severe Obese 2 (8.3%) 6 (25.0%) 2 (11.1%) 10 (55.6%)

References:

Kuczmarski RJ, Ogden CL, Guo SS, Grummer-Strawn LM, Flegal KM, Mei Z, Wei R, Curtin LR, Roche AF, Jhonson CL 2000. CDC Growth Charts for the United States: Methods and Development. Vital Health Stat. 2002; 11 246: 1-190.

Barlow SE, Expert Committee. Expert committee recommendations regarding the prevention, assessment, and treatment of child and adolescent overweight and obesity: summary report. Pediatrics. 2007; 120: Suppl 4:S164-92.

6. In the lines 296-298, the authors state that the study showed a significant reduction in BMD at different sites. It is not clear what the data were. The authors should clarify and show BMD data to support the statement.

Answer: We apologize and have changed Table 1 to present the data. 

7. Lines 312-319: The authors have an interesting finding pertaining to the sex differences, they speculate that it may be attributable to the differences in biochemical and anthropometric characteristics. It is not clear what biochemical characteristics are being referred to. Similarly, line 322-323, states males were much more compromised metabolically, however, it is not clear in which sense and what metabolic derangements this assessment is being referred to.

Answer: Dear Reviewer, we are referring to each of the nutritional and biochemical criteria used for the diagnosis of MeTS, in this case among the biochemicals HDLc, triglycerides, and glycemia. Among girls without MeTS, we were able to match with a group in which few biochemical/metabolic alterations were detected in 42% of them. 

In the group without the presence of Metabolic Syndrome, 42% of the patients presented no biochemical alterations, and 45.9% demonstrated weight (kg) and BMI (kg/m²) between ≥85th percentile and <95th percentile when compared to those with the presence of MetS where only 16.7% were between these values.

In contrast, for the boys:

On the other hand, in the group of adolescent males, it was observed, even among those without MetS, that 83% presented biochemical alterations (HDLc, triglycerides, and glycemia) and weight (kg) and BMI (kg/m2) higher or equal to the 95th percentile were observed in 55.5% of them. This confirms that even those without MeTS were much more compromised metabolically and nutritionally than the female adolescents.

8. In the conclusion session, authors mention the reduction in BMD and its effects on peak bone mass, however, there are no data presented on BMD values or Z-scores in this cohort.

Answer: We apologize again that we did not present the data. We have now added the missing information to Table 1. 

9.MetS increases bone resorption, however, CTX was lower in MetS group indicating lower bone resorption. It is eluded that most likely due to decrease bone remodeling. However, in chronic inflammation states of MetS and obesity osteoclast and bone resorption activities are significantly elevated as been shown in other adults studies and sites by the authors. In addition, authors showed that BMD negatively correlated with CTX. Can authors elaborate further on these rather unexpected findings?

Answer: We also believed that bone resorption should have increased, reflected by an increase in S-CTX, as documented in studies that evaluated adults. However, we should remember that adolescence is a time for increased bone mass and peak bone mass, as close to 42% of bone mass is added during the second decade of life. Therefore, bone formation should overcome resorption. Herein, we did not observe evolution as shown in adults. There was a reduction in the concentrations of all bone markers analyzed in both sexes among adolescents with MetS, suggesting an aggression to bone formation and resorption, possibly indicating a reduction in remodeling. 

10.Table 4 appears to have a small shift in the values, which should be an easy edit.

Answer: Thank you, we have corrected this.

11.Table 5 is difficult to read, p values are missing in the comparison of Lumbar spine BMD with the bone makers; instead, r values are provided twice.

Answer: We appreciate your careful evaluation, and have made the corrections.

12.Table 6 is difficult to read and follow, recommend revision. There are multiple comparisons in the study, especially in table 6. It is not clear if correction for multiple comparisons was done. Can authors comment?

Answer: Dear reviewer, this is now Table 5, and it is possible to observe the linear correlations (Pearson) between bone biomarkers and the components of the Syndrome. The correlations demonstrate that each Mets component in isolation does not have a significant correlation with bone biomarkers. Thus, the few significant p values have now been highlighted in bold and with an asterisk. 

13.Comment 3: Besides tables and results in the manuscript no additional data were provided. BMD data were not provided.

Answer: Thank you, we have made the corrections.

14.Comment 4: Discussion had comprehensive overview and review of available data on this and relevant topics. Many sentences are too long and difficult to follow. Discussion should be revised and edited for clarity.

Answer: We appreciate the considerations and the discussion has been revised and edited.

Line 59: "non-definitively demonstrated" - consider revision

Answer: We appreciate the comment. In the restructuring of the introduction this paragraph was removed. 

14. Sentence in lines 261-268 is not clear and seems ambiguous. Please revise the point.

Answer: We accept your suggestion about ambiguity and have restructured the text.

15.In line 268 authors use BMTs, I am assuming it's a typo and they mean BMD. This should be clarified and corrected. There was another area where BMTs were used in place of intended BMD.

Answer: We apologize, but BMTs were used as an abbreviation for Bone Turnover Markers. However, in the current version the problem does not arise.

16.Lines 303-305: the sentence is incomplete. this needs a grammar check.

Answer: You are correct, we have corrected the paragraph.

17.The statements in lines 332-338 need references. The sentence in lines 417-424 is confusing and very long, needs revision. 

Answer: Dear reviewer, we have added the respective references to lines 320-322 (Fernandes, Gonçalves, Brito, 2017; Cheung et al., 2013). In the other section, lines 352-355 have been reformulated: “However, in the present study there was a reduction in the concentrations of all bone markers analyzed in both sexes among adolescents with MetS, suggesting an aggression to bone formation and resorption, possibly indicating a reduction in remodeling”.

References:

Fernandes TAP, Gonçalves LML, Brito JAA. Relationships between Bone Turnover and Energy Metabolism. Journal of Diabetes Research.2017; 27: 1.

Cheung CL, Tan KC, Lam KS, Cheung BM. The relationship between glucose metabolism, metabolic syndrome, and bone-specific alkaline phosphatase: a structural equation modeling approach. J Clin Endocrinol Metab. 2013.9:3856-63.

Reviewer #2: General Comments

18.This appears to be a sub-study of a larger study looking at the impact of metabolic syndrome and its components on bone remodeling in adolescents. The authors present relevant data and the use of biochemical markers of bone turnover is appreciated given the age group and lack of research using bone markers in adolescents particularly in those who are classified as overweight or obese. This data would be a welcome addition to the literature.

Answer: Dear Reviewer, we appreciate your positive comments regarding our study.

20.There are number of general and specific comments that need to be addressed with respect to the interpretation of the bone remodeling data, editing of wording, and the discussion as it is currently written. For these reasons, a major revision is recommended.

As well, the Authors have a paper published in Bone (2014) examining MetS and its effects on bone mass, similarly using 271 participants and study design. Although that paper did not measure bone markers, it is not included in the current reference list even though sections of the methods could refer to it and make it clear if the 271 participants are similar between both papers. It appears the present study is a subset or additional analysis of a larger one that may have already been published. Are the current 84 participants from the original 271 participants or have any of them been additionally recruited? Can the authors provide any information or discussion on the use of or lack thereof a control group (normal weight non-MetS).

Answer: We apologize if it seemed that we had given prominence to our study, published in Bone (2014), as it appears among other references, in the Introduction (Nóbrega da Silva et al., 2014). There are few studies that address the theme in this age group and the current work is the subset of a larger work previously published, where we only analyzed the involvement of bone mineral density among adolescents with MetS. With respect to the inclusion of a control group with normal weight, it did not seem appropriate to conduct this study, considering the factors that could influence bone markers, when individuals are overweight and have MetS. In a previous publication (Mosca et al., 2017), we used a control group with adequate weight, when analyzing bone markers in obese individuals, without classifying them as MetS + or MetS –.

References:

Nóbrega da Silva V, Goldberg TBL, Mosca LN, Rizzo ACB, Teixeira AS, Corrente JE. Metabolic syndrome reduces bone mineral density in overweight adolescents. Bone. 2014; 66:1-7.

Mosca LN, Goldberg TBL, Silva VN, Kurokawa CS, Rizzo ACB, da Silva CC, Teixeira AS, Corrente JE. The impact of excess body fat on bone remodeling in adolescentes. Osteoporos Int. 2017. 28:1053–1062.

21.Much of the discussion is spent on studies and finding relating to OC, however that is not the only bone marker that was measured in the current study. CTx was mentioned briefly and there was no discussion provided to BAP. In addition, the authors provide meaningful information on types of diet, inflammation and cytokines in the discussion, but fail to use this information to support and make connections to the results of their study. Given that inflammatory cytokines or RANKL were not measured in the current study, I would recommend the authors not include these sections, unless they can connect it with the variables they did in fact measure. I would suggest speaking more to the specific MetS components, the number of them present and addressing S-CTx and BAP in the discussion.

Answer: We appreciate your suggestions, and have made adjustments to the text. 

Specific Comments Abstract

21.1. Numerical values or results in terms of r- or p-values should be presented in the abstract.

Answer: Dear reviewer, we accept your suggestion and the abstract now contains the r and p-values. 

Introduction

22.1. Define MetS, the conditions it can include, risk factors and how this relates to and presents in adolescents. How many MetS components do adolescents need to be classified as having MetS?

Answer: Dear Reviewer, the concept of MeTS and the recommendations proposed by the IDF, now read as follows in the methods (lines 160 to 165): “MetS was defined according to the criteria proposed by the IDF [Zimmet et al., 2007]. A subject was classified as having MetS if he/she presented central obesity, defined by a large waist circumference, and at least two of the four criteria shown in Table 1.” In the new version, Table 1 was excluded (on request) and the text containing the explanations has been added to the manuscript.

Reference:

Zimmet P, Alberti KG, Kaufman F, Tajima N, Silink M, Arslanian S, Wong G, Bennett P, Shaw P, Caprio S, IDF CONSENSUS GROUP (2007). The metabolic syndrome in children and adolescents–an IDF consensus report. Pediatr Diabetes 8:299-306.

23.2. In the second paragraph of the introduction (lines 62-69), make reference to bone development papers. Only one reference is used at the end of the paragraph.

Answer: You are absolutely right. In the new version of the manuscript a wide review of the introduction has been carried out and some paragraphs have been adjusted with inclusions and removals. 

24.3. Line 66, the use of the word “intense” to describe the imbalance between bone formation vs resorption due to growth should be rephrased.

Answer: Thank you for the suggestion, the alterations have been made.

25.4. Line 67-68, “under normal physiological conditions” and “dependent phenomena”, is this in reference to growth in adolescents or a healthy adult? How bone remodeling is occurring during adolescence as a result of growth and maturation needs to be considered when describing this process.

Answer: Thank you for the suggestion, the alterations have been made.

26.5. Line 74 mentions a systematic review, please provide reference.

Answer: We have added the reference [3], thank you.

Reference:

Silva VND, Fiorelli LNM, Silva CC, Kurokawa CS, Goldberg TBL. Do Metabolic syndrome and its components have an impact on bone mineral density in adolescents? Nutrition Metabolism (Lond). 2017; 14:1-7.

27.6. Line 85, what is meant by “specialized literature”?

Answer: Thank you for the suggestion. We have rewritten the sentence to add clarity.

28.7. It is mentioned, line 80, that components of MetS are negatively correlated with BMD in adolescents. It may be worth including and referencing literature that has demonstrate negative impacts of excess weight in adolescents on BMD, BMC and/or bone strength, in addition to the incidence of increased fracture. There is some literature that has demonstrated negative correlations between BMI or excess weight on marks of bone turnover. This can and should be used to support some of the findings related to MetS and bone properties.

Answer: Dear reviewer, we appreciate your comment. Our group is dedicated to understanding bone markers during adolescence and their relationship with excess weight, which has resulted in some publications. 

References:

Nóbrega da Silva V, Goldberg TBL, Mosca LN, Rizzo ACB, Teixeira AS, Corrente JE. Metabolic syndrome reduces bone mineral density in overweight adolescents. Bone. 2014; 66:1-7. 

Mosca LN, Goldberg TBL, Silva VN, Kurokawa CS, Rizzo ACB, da Silva CC, Teixeira AS, Corrente JE. The impact of excess body fat on bone remodeling in adolescentes. Osteoporos Int. 2017. 28:1053–1062.

Silva VND, Fiorelli LNM, Silva CC, Kurokawa CS, Goldberg TBL. Do Metabolic syndrome and its components have an impact on bone mineral density in adolescents? Nutrition Metabolism (Lond). 2017; 14:1-7.

The impact of excess body fat on bone remodeling was evaluated in overweight, obese, and extremely obese adolescents. In adolescents with excess weight, it was observed that the higher the bone mineral content and bone mineral density values, the lower the levels of the biomarkers. Nutritional imbalances through excess had a negative effect on bone formation in this stage of life. In girls with excess weight, the biomarkers were higher in the 10 to 13-year age group and no significant differences were observed between groups according to nutritional status. In boys, the levels were higher in those aged 13 to 15 years. According to nutritional status, significant differences were only observed in mean S-CTx for the age groups of 10–15 years, with higher levels between overweight and obese adolescents aged 10–12 years and between obese and extremely obese adolescents aged 13–15 years. In girls, significant negative correlations were observed between lean mass, fat mass, and fat percentage and each of the three bone markers studied. There was no correlation between lean mass or fat mass and the three biomarkers in boys. The biomarker trends demonstrated across the age groups follow the age trends for growth velocity. In conclusion, the higher the fat percentage and fat mass in girls, the lower the levels of the biomarkers, indicating that excess body fat has a negative effect on the evolution of these markers during adolescence. (Mosca et al., 2017).

Our research group has been investigating explanations about biomarkers and bone remodeling in adolescents with and without MetS, especially considering the lack of data in the literature on this population of adolescents.

29.1. Please refer to general comment as it pertains to previously published paper that could be referenced in this section of the document.

Answer: Dear reviewer, we have added our previous studies, thank you. 

References:

Nóbrega da Silva V, Goldberg TBL, Mosca LN, Rizzo ACB, Teixeira AS, Corrente JE. Metabolic syndrome reduces bone mineral density in overweight adolescents. Bone. 2014; 66:1-7. 

Mosca LN, Goldberg TBL, Silva VN, Kurokawa CS, Rizzo ACB, da Silva CC, Teixeira AS, Corrente JE. The impact of excess body fat on bone remodeling in adolescentes. Osteoporos Int. 2017. 28:1053–1062.

Silva VND, Fiorelli LNM, Silva CC, Kurokawa CS, Goldberg TBL. Do Metabolic syndrome and its components have an impact on bone mineral density in adolescents? Nutrition Metabolism (Lond). 2017; 14:1-7.

30.2. Final paragraph of section 2.1 (lines 132-136) describe how participants were matched and compare to the 271 subjects. Consider moving this to the first paragraph of section 2.1 to make it clear earlier in the paper how participants were matched for the study analysis.

Answer: Dear reviewer, we appreciate your suggestion, which was fully accepted.

31.3. Why did the authors choose to match participants by chronological age, bone age, and pubertal stage? Please provide reference or justification for using these matching criteria and perhaps for not including anthropometric measures.

Answer: Dear reviewer, the choice to match individuals by these variables resulted from the great differences observed between pubescent children in different stages of growth and physical maturity (early, middle, and late puberty). In addition, regarding the definition of metabolic syndrome in adolescents according to International Diabetes Federation criteria, pubescents aged 10 to 16 are analyzed as a single group, with only the percentile of waist circumference varying and the cutoff points for the other biochemical markers being similar for all criteria. In addition, from a statistical point of view, non-matching could generate bias in the explanations about the MetS outcome itself if the sample was not minimally homogenized.

32.4. Are the authors aware and able to report how many of the female participants were pre vs. post-menarcheal, or had all female participants reached menarche?

Answer: Dear reviewer, each individual among these 271 adolescents was enrolled and followed up at our Adolescence Service. So we can assure you that, as explained in the text, they were included in the sample with care.

As more than 60% of the adolescents, both those in the MetS + and MetS- groups, were in late puberty, we can suggest that they had already menstruated (see Table 1 in the new version). However, we do not have these data to confirm our assumption, as it would be necessary to ask each individual included.

33.5. It is not clear how the information that was used for the inclusion criteria was collected. Was this done by interview, survey, or a specific questionnaire. Specifically, how was the nutritional information collected?

Answer: Dear Reviewer, I believe that the answer to the previous question will resolve any doubts. In the text we have written (line 94): “on the occasion of their first visit to the Adolescent Unit of the Botucatu Medical School University Hospital (SP, Brazil), between 2011 and 2013”.

34.6. Line 141, potential editing error in sentence – “utilized to classify nutritional status”.

Answer: Dear reviewer, we accept your suggestion and have edited the sentence. 

35.7. Like 143 makes refence to growth curves. Are these based on WHO or CDC curves?

Answer: They are based on CDC growth charts.

Reference:

Kuczmarski RJ, Ogden CL, Guo SS, Grummer-475 Strawn LM, Flegal KM, Mei Z, Wei R, Curtin LR, Roche AF, Jhonson CL 2000. CDC Growth Charts for the United States: Methods and Development. Vital Health Stat. 2002; 11 246: 1-190.

Statistical Analysis/Results

36.1. Given that DXA was used, is it possible for the authors to provide data on lean and fat mass, or as a percentage, for the participants. Is there BMC data that can be reported?

Answer: Dear Reviewer, we have all these data in our parent spreadsheet. However, BMC data was also not presented in the study published in Bone (Nóbrega da Silva et al., 2014), as we have already commented to Reviewer 1. In that study, the analysis included all 271 adolescents, as shown in the table below.

Regarding fat mass and percentage, we have already published the association with bone markers, identifying a negative association, indicating that the greater the fat mass or its percentage, the lower the results of the concentrations of the markers. In the conclusion of this paper we have written: The higher the fat percentage and fat mass in girls, the lower the levels of the biomarkers, indicating that excess body fat has a negative effect on the evolution of these markers during adolescence.

References:

Nóbrega da Silva V, Goldberg TBL, Mosca LN, Rizzo ACB, Teixeira AS, Corrente JE. Metabolic syndrome reduces bone mineral density in overweight adolescents. Bone. 2014; 66:1-7.

Mosca LN, Goldberg TBL, Silva VN, Kurokawa CS, Rizzo ACB, da Silva CC, Teixeira AS, Corrente JE. The impact of excess body fat on bone remodeling in adolescents. Osteoporos Int. 2017. 28:1053–1062.

37.2. Table 2, the n-values for the Tanner stages do not add up to be the n-values for the respective group.

Answer: You are right, in some teenagers we did not obtain Tanner stages, however, we obtained bone age from all of them and we know that there is a strong correlation between these events.

38.3. It is not clear what Tanner stages the authors are using to define participants as early, mid or late puberty. Please define this in the maturation section of the methods.

Answer: We used the classification proposed by Marshall & Tanner, however, for the number of adolescents in each group the following were adopted, Breast 1- 2 considered early puberty, Breast 3 mid puberty, and Breast 4 and 5 late puberty, as well as in genitals in male adolescents (Table 1 in this new version of the manuscript).

Reference:

Finkelstein JW. The endocrinology of the adolescent. Symposium of adolescent medicine. Pediatrics Clinic of North America.1980; 27:53-69.

39.4. Statistical analysis section states (Line 215-218) than an ANOVA was conducted to compare bone markers with the number of MetS components, however results are not presented in the document.

Answer: Dear reviewer, you are absolutely right, we have removed this analysis from our study and present it in the new version of the manuscript.

40.5. In terms of analysis, the age range (10-16yrs) in addition to the inclusion of all tanner stages into the analysis may be problematic and contributing to some of the correlations and findings.

Answer: Dear reviewer, we agree and also considered this a problematic situation when we outlined the study. However, the IDF uses this group of individuals to perform the classification of those with or without MetS, in addition to the matching that took into account bone and chronological age and the pubertal stages. 

41. A) Bone marker concentrations change with maturity and there have been studies to show varying levels by tanner stage. I would expect to see negative correlations since BMD increases and bone marker concentrations decrease with age/maturation. It is not clear from the data and subsequent discussion why this relationship is significant in females and more so in those that have MetS. Is the number of MetS components, or a specific component, or perhaps their excess weight (percent body fat) a factor? Is it related to their maturation? Looking at the data more females are in late puberty compared to boys. Is the menarcheal status between the female MetS and non-MetS participants similar?

Answer: Dear Reviewer, all your questions are highly thought-provoking and we reflected on a number of them while writing our discussion. Our research group started on the topic by analyzing bone density and mineral content and bone markers in eutrophic adolescents, male and female. Next, we analyzed the effects of obesity on bone density and bone markers in adolescents. Thus, we are aware of this negative relationship that the reviewer so adequately points out (references already presented). However, in order to avoid doubts that the age effect alone was responsible for the reduction in bone markers evidenced in the current study, we present in the discussion the following extract (line 261- 276):

“Researchers have highlighted the importance of understanding the evolution of BMD and alterations in the levels of bone biomarkers during adolescence (Silva et al., 2012; Fortes et al., 2014). Fortes et al. (2014), evaluating BMD and its relationship with bone markers in a sample of 101 eutrophic female adolescents of 9 to 20 incomplete years, observed an expressive increase in BMD according to the progression of chronological and bone age. With respect to indicators of bone metabolism, higher mean biomarker levels (BAP=110.12 U/L, OC=26.49ng/mL, S-CTx=1.58 ng/mL) were found when the adolescents were in stage 3 of breast development and in early adolescence (11 to 12 years). The levels of bone biomarkers were lower at the end of adolescence (BAP=31.70 U/L, OC=4.98ng/mL, S-CTx=0.44 ng/mL). Similar results were recently published with the follow-up of 2,416 children and adolescents, from 3 months to 17 years of age, analyzing bone markers Osteocalcin, PINP, and CTX-I, 317 with obesity and the other eutrophic (Geserick et al., 2020). During puberty, the authors demonstrated that the highest levels of bone markers occurred from 10 to 11 years (girls, Tanner 3) and 13 years (boys, Tanner 3–4). Results very close to those presented by Fortes et al. (2014) regarding age and pubertal stage. 

In addition, our current group of teenagers included were between 10-16 years old, with an average CA of around 12 years for girls and 13 years for boys and BA of around 13 years for both.

(Line: 277 – 289): “In male adolescents, Silva et al. (2012) studied a sample of 87 eutrophic adolescents aged 10 to 18 years and concluded that serum biomarker levels differed significantly among age groups. The highest concentration of the biomarkers was observed between 13 and 15 years of age (BAP = 155.50U/L, OC = 41.63 ng/mL, S-CTx = 2.09 ng/mL) and the lowest concentration between 16 and 18 years (BAP= 79.80U/L, OC = 27.80 ng/mL, S-CTx = 1.60 ng/mL). Within this context, proportionally higher levels of bone biomarkers would be expected in the adolescents in this present study, since the mean CA of girls with or without MetS was around 12 years of age and BA, around 13 years of age, while in boys both CA and BA were around 13 years of age. The concentrations obtained among adolescents without MetS in the present study were close to those observed among the eutrophic individuals analyzed in the studies cited (Silva et al., 2012; Fortes et al., 2014). However, a significant reduction in both biomarkers of bone formation and resorption was observed in the group of adolescents with MetS compared to the matched group without MetS”.

With respect to the negative correlations, this was part of one of the responses to the Editor (Comment 3):

References:

Silva CC, Kurokawa CS, Nga HS, Moretto MR, Dalmas JC, Goldberg TBL. Bone metabolism biomarkers, body weight, and bone age in healthy Brazilian male adolescents. J Pediatr Endocrinol Metab. 2012; 25: 479-484.

Fortes CMT, Goldberg TBL, Kurokawa CS, Silva CC, Moretto MR, Biason TP, Teixeira AS, Nunes HRC. Relação entre as idades cronológica e óssea e o estágio puberal das mamas com os biomarcadores ósseos e a densidade mineral óssea em adolescentes. J Pediatr (Rio J). 2014; 90:624-631.

Geserick M, Vogel M, Eckelt F, Schlingmann M, Hiemisch A, Baber R, Thiery J, Korner A, Kiess W, Kratzsch J. Children and adolescents with obesity have reduced serum bone turnover markers and 25-hydroxyvitamin D but increased parathyroid hormone concentrations - Results derived from new pediatric reference ranges. Bone. 2020;132:115-124.

42. B) Because of the changing bone marker concentrations with maturation it is not uncommon for studies to separate participants into groups based on tanner staging. Have the authors considered creating 2 tanner groups to help minimize maturation effects that come with having all tanners 1-5 grouped together?

Answer: Dear reviewer, we understand your concern and believe that our comments on the previous questions demonstrate why we did not perform the analysis based on pubertal stages. In addition, there is no recommendation for the classification of MetS by pubertal stages in adolescents, the 10-16 age group being considered as a single group (IDF, 2007).

Reference:

Zimmet P, Alberti KG, Kaufman F, Tajima N, Silink M, Arslanian S, Wong G, Bennett P, Shaw P, Caprio S, IDF CONSENSUS GROUP (2007). The metabolic syndrome in children and adolescents–an IDF consensus report. Pediatr Diabetes 8:299-306.

43.C) As well, given that BMD is affected by body size, have the authors considering or tried adjusting their analysis by controlling for various anthropometric measures (i.e. height)?

Answer: Dear reviewer, reviewer 1 asked a similar question. We ask you to please consider the answer to question 4.

Discussion

44.1. Line 256 – in reference to the “behavior”’ of bone biomarkers. Please rephrase to be clearer as to what the results were.

Answer: Dear reviewer, we have made the adjustments in the new version of the manuscript. 

45.2. Line 259 – states that BMD was negatively correlated with all bone markers in adolescents with MetS. However, this was only the case with females and not males. Please correct this statement.

Answer: Dear reviewer, you are right, we have added that the correlations were negative only in girls (First paragraph of the discussion).

46.3. Lines 261-268 – authors discuss collinearity of the bone markers. What is the purpose of this finding and how does it relate to the literature and data?

Answer: Dear reviewer, you are right, we have deleted this paragraph.

47. 4. Line 268 – what is BMT?

Answer: Dear reviewer, Bone markers of turnover (BMT). We are sorry we did not write this in full.

48. 5. Provide reference at end of sentence (line 271)

Answer: Dear reviewer, we have added the reference [30] to the end of the sentence in this new version of the manuscript. 

Reference:

Weaver CM, Gordon CM, Janz KF, Kalkwarf HJ, Lappe JM, Lewis R, Okarma M, Wallace TC, Zemel BS. The national osteoporosis foundation's position statement on peak bone mass development and lifestyle factors: a systematic review and implementation recommendations. Osteoporos Int. 2016; 27:1281-1386.

49. 6. Paragraph (Line 277-295) – Authors discuss bone marker levels by age and tanner stage. Please be specific if you are referring to males vs. females in those studies. And which of these studies used participants with MetS. As well, how does these studies and their grouping compare given that the current study includes all tanner stages into one group. Please refer to comment 5b in previous section.

Answer: Dear reviewer, the answers to this comment can be found in other questions (41A and 42B). We emphasize that the studies cited were developed for eutrophic girls and boys, independently. We did not detect articles that evaluate bone turnover markers in adolescents with MetS in the literature. Some authors have analyzed Osteocalcin and its relationship with some MetS components, and these are mentioned in the discussion. 

50.7. Line 307 – Unclear what the results of Tan et al were and how they compare to the current findings. Please elaborate.

Answer: Dear reviewer, this paragraph has been reviewed and adjusted, highlighting the negative correlation between OC and MetS that occurred for males. Tan et al. (2011) evaluated 2,344 Chinese men between 20 and 69 years of age and found a negative association between OC levels and the presence of MetS. These results agree with our findings, however, in our results the reductions in OC were significant and negative only in female adolescents with MetS. 

Reference:

Tan A, Gao Y, Yang X, Zhang H, Qin X, Mo L, Peng T, Ning X, Mo Z. Low serum osteocalcin level is a potential marker for metabolic syndrome: results from a Chinese male population survey. Metabolism. 2011; 60:1186-1192.

51. 8. Line 318 – what do the authors mean by “biochemical alterations”?

Answer: We ask the Reviewer to please note the response to Comment 7 to Reviewer 1.

52.9. Paragraph (lines 324-327) – small paragraph talking about OC. Consider reorganizing and placing it within previous paragraph where authors discuss OC. As well, please provide clearer discussion of findings when comparing current study results with study using eutrophic children, when such a group was not used in the current study.

Answer: Dear Reviewer, we appreciate your suggestion and have deleted the paragraph, as the comparison was actually made with eutrophic children.

53.10. Lines 334-335 – please provide references

Answer: Dear Reviewer, we agree and have reorganized the entire paragraph including references.

54.11. It is not clear the connection the authors are trying to make between different types of OC and various other physiological markers, and how this relates to the finding of the current study.

Answer: Dear Reviewer, we agree and have reorganized the text of the discussion in this new version of the manuscript.

55.12. Please provide discussion on limitations for the current study. Please include some methodological limitations of using DXA measurement with children and adolescents.

Answer: Dear Reviewer, we have included the study limitations at the end of the discussion section. Details of the evaluation of bone mineral density are highlighted in the methods section.

56.13. Line 426 and 427 – the authors mention “raising new hypotheses” and suggesting possible “design for future studies”, although hypotheses were not presented within the document. Can the authors please include some examples for potential future studies or considerations.

Answer: Dear Reviewer, we have included this information in the new version of the manuscript. Thank you again for the detail of the review and we hope to have met your expectations.

---

## [Decision Letter · Decision Letter 1]

18 Feb 2021

PONE-D-20-24612R1

Impact of metabolic syndrome and its components on bone remodeling in adolescents

PLOS ONE

Dear Dr. Goldberg,

Thank you for submitting your revised manuscript to PLOS ONE. After careful consideration, we feel that it has merit but does not fully meet PLOS ONE’s publication criteria as it currently stands. Therefore, we invite you to submit a revised version of the manuscript that addresses the points raised during the review process.

Please carefully review the comments below and address each one in a comprehensive manner.

Ensure that height, weight, and BMI are expressed as z-scores in your Table 1 and other appropriate areas.

Please ensure that your BMD data are also reported as z-scores which is the standard method or reporting in children and adolescents. This should be easy to extract from the radiological reports

We look forward to receiving your revised manuscript.

Kind regards,

Benjamin Udoka Nwosu, MD

Academic Editor

PLOS ONE

Reviewers' comments:

Reviewer's Responses to Questions

**Comments to the Author**

1. If the authors have adequately addressed your comments raised in a previous round of review and you feel that this manuscript is now acceptable for publication, you may indicate that here to bypass the “Comments to the Author” section, enter your conflict of interest statement in the “Confidential to Editor” section, and submit your "Accept" recommendation.

Reviewer #1: (No Response)

2. Is the manuscript technically sound, and do the data support the conclusions?

Reviewer #1: Partly

3. Has the statistical analysis been performed appropriately and rigorously? 

Reviewer #1: N/A

4. Have the authors made all data underlying the findings in their manuscript fully available?

Reviewer #1: Yes

5. Is the manuscript presented in an intelligible fashion and written in standard English?

Reviewer #1: Yes

6. Review Comments to the Author

Reviewer #1: The authors made major revisions to the manuscript after the first round of reviews. The manuscript is clearer and presents major findings. I would suggest graphs for some of the findings vs. tables.

My comments are incorporated within the comment session and the manuscript in red and are attached to this review.

In brief:

Abstract: line 48 - consider revision

Methods:

lines 166-167: could authors comment on anyone with impaired glucose tolerance and/or T2DM? This cohort present a more significant metabolic milieu and would be interesting to see the effects on bone and may need to be separated from non-DM. This should be clarified for both cohorts (MetS - and MetS+).

session 2.5 of Methods: BMD is not height adjusted, though groups didn't differ in height. I would include and explain this in the methods.

Results:

Also, I would suggest to the authors to include and do analysis of BMD z-scores as BMD done in pediatric population. This is a major point.

Table 1 and results section discussing Tanner staging: The authors included Tanner stage 1 as early puberty. Tanner 1 is pre-pubertal stage. I suggest authors to correct this and separate subjects with Tanner stage 1 if there were any.

Discussion:

Much of the beginning of the discussion is spent on age and puberty affects on bone mass accrual. However, the study's focus is a cross-sectional assessment of MetS on BMD and bone remodeling of age matched groups. Though the groups are heterogeneous incorporating wide age range (10-16 years) and all pubertal stages, there were no clear analysis of affects of age and puberty on the aims of the study in the current study. The first part of the discussion should be revised and shortened to tie in with the findings and the focus of the study.

Also, discussion is focused mainly around bone remodeling markers. There is no discussion of the other findings such as BMD, opposite findings in correlation analyses (Table 3-5), which need to be explained and elaborated on. I suggest major revisions to the discussion session.

Line 256: states at the beginning of puberty, which is not entirely correct. In girls, growth velocity peak is late Tanner 2 to 3 of breast development. In boys, growth spurt is mainly TS 3, occasionally to TS 4, which is mid to later puberty. Similarly, bone mass accrual happens later in boys than girls. The authors also define TS 3 as mid puberty, thus, the statement in line 256 should be corrected.

Line 260-261: Authors refer to international recommendations, but no recommendations are provided or sited. It is not clear what authors are referring to.

Paragraph lines 292-294: is out of place as the discussion in preceding and subsequent paragraphs is about biochemical bone remodeling markers. Though, BMD should be discussed as in earlier suggestion.

Please see other comments and questions within the discussion session of the manuscript: highlighted and annotated in red.

7. PLOS authors have the option to publish the peer review history of their article (what does this mean?). If published, this will include your full peer review and any attached files.

Reviewer #1: No

---

## [Author Response · Author response to Decision Letter 1]

27 Mar 2021

PONE-D-20-24612R1

Impact of metabolic syndrome and its components on bone remodeling in adolescents

PLOS ONE

To Benjamin Udoka Nwosu, MD

Academic Editor

PLOS ONE

Dear Professor Udoka Nwosu 

First, we are grateful for the opportunity to send a revised version of our paper and for the excellent comments from the reviewer regarding our article “Impact of metabolic syndrome and its components on bone remodeling in adolescents”

Editor Comments

Please carefully review the comments below and address each one in a comprehensive manner.

Ensure that height, weight, and BMI are expressed as z-scores in your Table 1 and other appropriate areas.

Dear Prof Udoka Nwosu, we thank you for the suggestion and have included the z-score for height, weight, and BMI.

Please ensure that your BMD data are also reported as z-scores which is the standard method or reporting in children and adolescents. This should be easy to extract from the radiological reports.

Regarding the z-scores of BMDs, we would ask you to understand the reasons we are unable to meet your request. The data referring to Densitometries were obtained between 2011 and 2012 and all anthropometric, biochemical, and densitometric variables were transferred to a spreadsheet. At that time, only the BMDs were transcribed and not the z-scores. The data for the 271 adolescents were published in another journal (BONE) in 2014. At this point, we would like to point out that the Densitometry machine has been deactivated and our University has acquired another machine, of another brand. In addition, the hospital has undergone a digitization process, which took place in 2012. Thus, printed data are extremely difficult to locate after all this time. We would be grateful for any guidance on your part on how to determine the z-score, and compare with which benchmarks, as we do not have Brazilian benchmarks for adolescent densitometry, in addition to the studies cited in the study. 

Review Comments to the Author

Reviewer #1: The authors made major revisions to the manuscript after the first round of reviews. The manuscript is clearer and presents major findings. I would suggest graphs for some of the findings vs. tables.

My comments are incorporated within the comment session and the manuscript in red and are attached to this review.

In brief:

Abstract: line 48 - consider revision

Thank you for the suggestion, the alterations have been made.

Methods:

lines 166-167: could authors comment on anyone with impaired glucose tolerance and/or T2DM? This cohort present a more significant metabolic milieu and would be interesting to see the effects on bone and may need to be separated from non-DM. This should be clarified for both cohorts (MetS - and MetS+).

Dear Reviewer, we know how frequent Type 2 Diabetes in the USA is among obese, severely obese, and MetS adolescents, but in our studies, including an article published in 2013, we report that for both sexes, altered glycemia was the least prevalent MetS factor among all factors analyzed. In the general sample of 271 adolescents evaluated, 2% of male adolescents and 3% of females presented glucose intolerance.

When we analyzed this factor in the group of 84 paired adolescents, in the MetS+ group, 3 of 24 female adolescents had glucose intolerance and none among the MetS-. For boys with MetS+, 2 out of 18 had glucose intolerance and none among the MetS-.

Given this information, we do not have enough numbers to analyze your important suggestion.

Reference

Rizzo ACB, Goldberg TBL, Silva CC, Kurokawa CS, Nunes HRC, Corrente JE.

Metabolic syndrome risk factors in overweight, obese, and extremely obese Brazilian adolescents. Nutrition journal 12 (1), 1-7. https://doi.org/10.1186/1475-2891-12-19

session 2.5 of Methods: BMD is not height adjusted, though groups didn't differ in height. I would include and explain this in the methods.

Results: Also, I would suggest to the authors to include and do analysis of BMD z-scores as BMD done in pediatric population. This is a major point.

Dear Reviewer,

We have answered Prof. Nwosu on this important consideration and the problems involved with the BMD z score.

“Regarding the z-scores of BMDs, we would ask you to understand the reasons we are unable to meet your request. The data referring to Densitometries were obtained between 2011 and 2012 and all anthropometric, biochemical, and densitometric variables were transferred to a spreadsheet. At that time, only the BMDs were transcribed and not the z-scores. The data for the 271 adolescents were published in another journal (BONE) in 2014. At this point, we would like to point out that the Densitometry machine has been deactivated and our University has acquired another machine, of another brand. In addition, the hospital has undergone a digitization process, which took place in 2012. Thus, printed data are extremely difficult to locate after all this time. We would be grateful for any guidance on your part on how to determine the z-score, and compare with which benchmarks, as we do not have Brazilian benchmarks for adolescent densitometry, in addition to the studies cited in the study.”

Table 1 and results section discussing Tanner staging: The authors included Tanner stage 1 as early puberty. Tanner 1 is pre-pubertal stage. I suggest authors to correct this and separate subjects with Tanner stage 1 if there were any.

Thank you for your important suggestion. The problem has been solved.

Discussion:

Much of the beginning of the discussion is spent on age and puberty affects on bone mass accrual. However, the study's focus is a cross-sectional assessment of MetS on BMD and bone remodeling of age matched groups. Though the groups are heterogeneous incorporating wide age range (10-16 years) and all pubertal stages, there were no clear analysis of affects of age and puberty on the aims of the study in the current study. The first part of the discussion should be revised and shortened to tie in with the findings and the focus of the study.

We appreciate your recommendations and have made the adjustments.

Also, discussion is focused mainly around bone remodeling markers. There is no discussion of the other findings such as BMD, opposite findings in correlation analyses (Table 3-5), which need to be explained and elaborated on. I suggest major revisions to the discussion session.

Line 256: states at the beginning of puberty, which is not entirely correct. In girls, growth velocity peak is late Tanner 2 to 3 of breast development. In boys, growth spurt is mainly TS 3, occasionally to TS 4, which is mid to later puberty. Similarly, bone mass accrual happens later in boys than girls. The authors also define TS 3 as mid puberty, thus, the statement in line 256 should be corrected.

We thank you for your comments and hope we have met your requests.

Line 256: states at the beginning of puberty, which is not entirely correct. In girls, growth velocity peak is late Tanner 2 to 3 of breast development. In boys, growth spurt is mainly TS 3, occasionally to TS 4, which is mid to later puberty. Similarly, bone mass accrual happens later in boys than girls. The authors also define TS 3 as mid puberty, thus, the statement in line 256 should be corrected.

We thank you for your comments and have made the necessary changes.

Line 260-261: Authors refer to international recommendations, but no recommendations are provided or sited. It is not clear what authors are referring to.

Paragraph lines 292-294: is out of place as the discussion in preceding and subsequent paragraphs is about biochemical bone remodeling markers. Though, BMD should be discussed as in earlier suggestion. Please see other comments and questions within the discussion session of the manuscript: highlighted and annotated in red.

Thank you for the generous and detailed review.

---

## [Decision Letter · Decision Letter 2]

23 Apr 2021

PONE-D-20-24612R2

Impact of metabolic syndrome and its components on bone remodeling in adolescents

PLOS ONE

Dear Dr. Goldberg,

Thank you for submitting your manuscript to PLOS ONE. After careful consideration, we feel that it has merit but does not fully meet PLOS ONE’s publication criteria as it currently stands. Therefore, we invite you to submit a revised version of the manuscript that addresses the points raised during the review process.

Please pay careful attention to the comments from the Reviewers. Ensure that all comments are addressed in full as this will be the final review process for your manuscript, and the Reviewers will make a final decision based on your response.

We look forward to receiving your revised manuscript.

Kind regards,

Benjamin Udoka Nwosu, MD

Academic Editor

PLOS ONE

Journal Requirements:

Reviewers' comments:

Reviewer's Responses to Questions

**Comments to the Author**

1. If the authors have adequately addressed your comments raised in a previous round of review and you feel that this manuscript is now acceptable for publication, you may indicate that here to bypass the “Comments to the Author” section, enter your conflict of interest statement in the “Confidential to Editor” section, and submit your "Accept" recommendation.

Reviewer #1: (No Response)

Reviewer #3: All comments have been addressed

2. Is the manuscript technically sound, and do the data support the conclusions?

Reviewer #1: (No Response)

Reviewer #3: Yes

3. Has the statistical analysis been performed appropriately and rigorously? 

Reviewer #1: (No Response)

Reviewer #3: Yes

4. Have the authors made all data underlying the findings in their manuscript fully available?

Reviewer #1: (No Response)

Reviewer #3: Yes

5. Is the manuscript presented in an intelligible fashion and written in standard English?

Reviewer #1: (No Response)

Reviewer #3: Yes

6. Review Comments to the Author

Reviewer #1: Author’s response to DXA z-score question:

Dear Reviewer,

We have answered Prof. Nwosu on this important consideration and the problems involved

with the BMD z score.

“Regarding the z-scores of BMDs, we would ask you to understand the reasons we are unable

to meet your request. The data referring to Densitometries were obtained between 2011 and

2012 and all anthropometric, biochemical, and densitometric variables were transferred to

a spreadsheet. At that time, only the BMDs were transcribed and not the z-scores. The data

for the 271 adolescents were published in another journal (BONE) in 2014. At this point, we

would like to point out that the Densitometry machine has been deactivated and our

University has acquired another machine, of another brand. In addition, the hospital has

undergone a digitization process, which took place in 2012. Thus, printed data are extremely

difficult to locate after all this time. We would be grateful for any guidance on your part on

how to determine the z-score, and compare with which benchmarks, as we do not have

Brazilian benchmarks for adolescent densitometry, in addition to the studies cited in the

study.”

Reviewer’s suggestions:

Lines 180 to 195: DXA methods: Please include a relevant line with the explanation of BMD DXA data and lack of Z-scores in methods as per authors explanation (see above).

Also, can consider extrapolating Z-scores from on-line calculator from CHOP (Zemel, et al. (2011). Revised Reference Curves for Bone Mineral Content and Areal Bone Mineral Density According to Age and Sex for Black and Non-Black Children:

Results of the Bone Mineral Density in Childhood Study. J Clin Endocrinol Metab, 96(10), 3160–3169. https://zscore.research.chop.edu/bmdCalculator.php ). if model of the machine used is the same. If deemed appropriate to include by the authors and editors, this would need to be clearly explained in the methods

Lines 221 to 227: Table 1 results: I suggest including the results of BMD from table 1 here as they are part of the aims.

Discussion:

Lines 253-259: It is not clear why BMD results in groups with and without MetS are not being discussed. Authors mention that they have previously reported BMD results. Assuming this study includes new results comparing the groups with and without MetS to evaluate the effects of MetS on BMD (Table 1), then they should be discussed in addition to MetS effects on bone remodeling that authors discuss well and in depths.

Authors aims:

“In view of the results already published [3,5,6], the objective of the current study was to evaluate the impact of MetS and each of its components on BMD and on biochemical markers of bone formation and resorption in adolescents.”

Line 271: reference of previously published data needed

Line 278: newly added discussion on CMR z-score. Please define/describe what it means.

Reviewer #3: This is a well written manuscript. The data on bone biomarkers is interesting.

Minor comments:

1. Lines 269- 271: Need reference for the transformed BMD values.

2. Table 1: Transformed BMD values are presented. Please describe the methodology for transforming BMD under statistical methodology.

3. Lines 42, 43: Reword as “The adolescents with excess weight and MetS exhibited a significantly lower transformed BMD and reduced concentrations of BAP, …

4. Ideally bone mineral content (BMC) (grams) and areal BMD (BMC/bone area g/cm2) are to be adjusted as whole body bone mineral apparent density (BMAD, g/cm3), calculated using the formula, BMC/ [whole body mineral area2/body height].  The authors state that both groups had similar heights. It is important to mention the limitations of not reporting the bone mineral apparent density(BMAD) -a size-adjusted measure of DXA BMD.

May add these references: Bachrach LK, Hastie T, Wang MC, Narasimhan B, Marcus R1999Bone mineral acquisition in healthy Asian, Hispanic, black, and Caucasian youth: a longitudinal study. J Clin Endocrinol Metab84:4702–4712

Height Adjustment in Assessing Dual Energy X-Ray Absorptiometry Measurements of Bone Mass and Density in Children; Babette S. Zemel, Mary B. Leonard, Andrea Kelly, Joan M. Lappe,Vicente Gilsanz, Sharon Oberfield, Soroosh Mahboubi, John A. Shepherd,Thomas N. Hangartner, Margaret M. Frederick, Karen K. Winer,and Heidi J. Kalkwarf

7. PLOS authors have the option to publish the peer review history of their article (what does this mean?). If published, this will include your full peer review and any attached files.

Reviewer #1: No

Reviewer #3: No

---

## [Author Response · Author response to Decision Letter 2]

9 Jun 2021

To Benjamin Udoka Nwosu, MD

Academic Editor

PLOS ONE

Dear Prof Udoka Nwoso,

We thank you for your important suggestions that helped improve our manuscript. 

Below please find our replies to the suggestions of the reviewers. We again thank the reviewers for their important comments and the detailed appraisal of our manuscript.

We hope to have clarified all doubts and remain at your disposal for further clarifications. 

Sincerely,

Prof. Tamara Goldberg and authors

Reviewers' comments:

Reviewer #1: Author’s response to DXA z-score question:

Answer:

Dear Reviewer,

We have answered Prof. Nwosu on this important consideration and the problems involved with the BMD z score.

“Regarding the z-scores of BMDs, we would ask you to understand the reasons we are unable to meet your request. The data referring to Densitometries were obtained between 2011 and 2012 and all anthropometric, biochemical, and densitometric variables were transferred to a spreadsheet. At that time, only the BMDs were transcribed and not the z-scores. The data for the 271 adolescents were published in another journal (BONE) in 2014. At this point, we would like to point out that the Densitometry machine has been deactivated and our University has acquired another machine, of another brand. In addition, the hospital has undergone a digitization process, which took place in 2012. Thus, printed data are extremely difficult to locate after all this time. We would be grateful for any guidance on your part on how to determine the z-score, and compare with which benchmarks, as we do not have Brazilian benchmarks for adolescent densitometry, in addition to the studies cited in the study.”

Reviewer’s suggestions:

Lines 180 to 195: DXA methods: Please include a relevant line with the explanation of BMD DXA data and lack of Z-scores in methods as per authors explanation (see above).

Also, can consider extrapolating Z-scores from on-line calculator from CHOP (Zemel, et al. (2011). Revised Reference Curves for Bone Mineral Content and Areal Bone Mineral Density According to Age and Sex for Black and Non-Black Children: Results of the Bone Mineral Density in Childhood Study. J Clin Endocrinol Metab, 96(10), 3160–3169. https://zscore.research.chop.edu/bmdCalculator.php ). if model of the machine used is the same. If deemed appropriate to include by the authors and editors, this would need to be clearly explained in the methods

Answer: Dear Reviewer, as we explained earlier, we could not redeem the BMD for age Z-score as our densitometry machine has been deactivated. Thank you for sending the references and for the availability of the calculator. Thus, we performed the BMD for age Z-score and, even by the equation proposed by Zemel et al. (2011), contained in the same calculator, from the study “Revised Reference Curves for Bone Mineral Content and Areal Bone Mineral Density According to Age and Sex for Black and Non-Black Children: Results of the Bone Mineral Density in Childhood Study. J Clin Endocrinol Metab, 96(10), 3160–3169, a BMD for height- for- age Z-score (HAZ).

As for the BMD for age Z-score, the results have been added to Table 1. As noted, they do not differ significantly according to sex between those with MetS– and MetS+. In addition, because the height between the matched groups does not differ statistically, as well as the height Z-score (Table 1), we performed the BMD calculations for height- for- age Z-score (HAZ) and, with the exception of the Lumbar BMD for height-for-age Z-score (HAZ), among boys, no other results differed significantly.

Just for your information, we have placed the results in the table below:

Table: Values of BMD for height- for- age Z-score (HAZ)

Values of BMD adjusted BMD Z-scores based on HAZ.

 Female (n=48) Male (n=36)

 MetS (-) (n=24) MetS (+) (n=24) MetS (-)

(n=18) MetS (+)

(n=18) 

 Mean ± SD P value Mean ± SD P value

Lumbar BMD 0.558 ± 1.00 0.547 ± 1.22 0.97 -0.511 ± 1.05 0.620 ± 0.88 0.049*

Total body BMD -0.665 ± 1.34 -0.377 ± 1.10 0.43 -0.740 ± 0.83 -0.628 ± 0.84 0.690

Subtotal BMD -0.684 ± 1.47 -0.223 ± 0.90 0.20 -0.601 ± 0.75 -0.474 ± 0.83 0.630

Lumbar Spine BMD for Age Z-Score adjusted for HAZ (height-for-age Z-score)

Total Body BMD for Age Z-Score adjusted for HAZ (height-for-age Z-score)

Total Body Less Head BMD for Age Z-Score adjusted for HAZ (height-for-age Z-score)

And in the study, we detail:

As our densitometry machine has been deactivated, it was impossible to retrieve the results of the BMD for age Z-score, in view of the long time since the evaluations were performed. Thus, we calculated the Lumbar BMD for age Z-score, Total body BMD, and Subtotal body BMD for age Z-score using the calculator https://zscore.research.chop.edu/bmdCalculator.php [28], to offer a guideline for readers. The effect of body size on bone mass can lead to misinterpretations when comparing individuals of different heights and body compositions. In the current study, the statures and height for age Z-scores did not differ between MetS + and MetS- groups for females (p=0.167; p=0.833) and males (p=0.728; p=0.881) respectively, and we sought to minimize body size-induced biases in bone mass estimates by using BMD transformed by body weight (g/cm²/kg body weight) [29]. This transformation can present limitations. Thus, authors use an estimate of volumetric bone mineral apparent density, BMAD (grams per cm3), published by Bachrach et al. [30] and adjusted by Zemel et al [31]. Zemel et al. [31] use a complex predictive equation of height-for-age Z-score (HAZ) (results not shown). All assessments were performed by only one technician trained for this purpose who had no knowledge of the research. In addition, the instructions of the manufacturer of the device and the standards recommended by the International Society for Clinical Densitometry [32] were followed.”

28. Zemel BS, Kalkwarf HJ, Gilsanz V, Lappe JM, Oberfield S, Shepherd JA, Frederick MM, Huang X, Lu M, Mahboubi S, Hangartner T, Winer KK. Revised reference curves for bone mineral content and areal bone mineral density according to age and sex for black and non-black children: results of the bone mineral density in childhood study. J Clin Endocrinol Metab. 2011 Oct;96(10):3160-9. 

30. Bachrach LK, Hastie T, Wang MC, Narasimhan B, Marcus R. Bone mineral acquisition in healthy Asian, Hispanic, black, and Caucasian youth: a longitudinal study. J Clin Endocrinol Metab.1999; 84:4702–4712.

31. Zemel BS, Leonard MB, Kelly A, Lappe JM, Gilsanz V, Oberfield S, Mahboubi S, Shepherd JA, Hangartner TN, Frederick MM, Winer KK, Kalkwarf HJ. Height adjustment in assessing dual energy x-ray absorptiometry measurements of bone mass and density in children. J Clin Endocrinol Metab. 2010; 95(3):1265-1273.

Lines 221 to 227: Table 1 results: I suggest including the results of BMD from table 1 here as they are part of the aims.

Answer: We appreciate your suggestion and have made the adjustments.

We have added in the manuscript: “The comparison of lumbar spine, proximal femur, and total and subtotal body BMD values according to sex and the presence of MetS revealed no significant differences. However, when BMD was transformed to BMD per kilogram of body weight, the female adolescents with MetS exhibited significant decreases in BMD at all sites evaluated (p<0,01) and the males for total and subtotal body BMD (p<0.05) (Table 1)”.

Discussion:

Lines 253-259: It is not clear why BMD results in groups with and without MetS are not being discussed. Authors mention that they have previously reported BMD results. Assuming this study includes new results comparing the groups with and without MetS to evaluate the effects of MetS on BMD (Table 1), then they should be discussed in addition to MetS effects on bone remodeling that authors discuss well and in depths.

Authors aims:

“In view of the results already published [3,5,6], the objective of the current study was to evaluate the impact of MetS on BMD and of MetS and each of its components and on biochemical markers of bone formation and resorption in adolescents.”

Answer: We are grateful for the suggestion and have adjusted the objectives and also added the information requested in the first paragraph of the discussion, between lines 253-259. 

In view of the results already published [3,5,6], the objective of the current study was to evaluate the impact of MetS on BMD, as well as the impact of MetS and each of its components on biochemical markers of bone formation and resorption in adolescents.

 add to first paragraph of the discussion…..

Our results showed that adolescents with excess weight and MetS exhibited reduced transformed BMD values at all sites for females, and total and subtotal body densities for males when compared to adolescents without MetS who were matched for chronological age, bone age, and pubertal stage for each sex.

With regard to the 3rd paragraph of the Discussion, we have added...

“We have previously reported that adolescents with excessive weight and MetS exhibited reduced transformed BMD values at all sites (Lumbar spine (L1-L4), proximal left femur, and total and subtotal) compared to adolescents without MetS [5] and the results of the current study corroborate those previously published. In a previous study [5] female adolescents with a large waist circumference…”

Line 271: reference of previously published data needed.

Answer: We thank you and have added the reference [5] in this line as well.

Line 278: newly added discussion on CMR z-score. Please define/describe what it means.

Answer: We thank you and have added the meaning of CMR z-score.

"Confirming our evidence, a recent study evaluated 306 female adolescents aged 9 to 12 years, analyzing the isolated and combined effect of cardiometabolic risk factors (CMR) (calculated by the CMR z-score of the included sample where CMR z-scores were created from the logarithmic transformed form of the variables: homeostatic model assessment of insulin resistance (HOMA-IR), C-reactive protein (CPR), triglycerides (TG), high-density lipoprotein cholesterol (HDL-c), low-density lipoprotein cholesterol (LDL-c), and mean arterial blood pressure (MAP)) on bone mass using DXA together with the evaluation of bone geometry, vBMD, and bone strength through the use of peripheral quantitative computed tomography (pQCT) in the tibia and femur.”.

Reviewer #3: This is a well written manuscript. The data on bone biomarkers is interesting.

Minor comments:

1. Lines 269- 271: Need reference for the transformed BMD values.

Answer: We appreciate your suggestion and have included the reference.

2. Table 1: Transformed BMD values are presented. Please describe the methodology for transforming BMD under statistical methodology.

Answer: We appreciate the suggestion and have placed the information where recommended (note: Table 1).

3. Lines 42, 43: Reword as “The adolescents with excess weight and MetS exhibited a significantly lower transformed BMD and reduced concentrations of BAP, …

Answer: Thank you very much for your suggestion.

4. Ideally bone mineral content (BMC) (grams) and areal BMD (BMC/bone area g/cm2) are to be adjusted as whole body bone mineral apparent density (BMAD, g/cm3), calculated using the formula, BMC/[whole body mineral area2/body height]. The authors state that both groups had similar heights. It is important to mention the limitations of not reporting the bone mineral apparent density(BMAD) -a size-adjusted measure of DXA BMD.

Answer: We appreciate your suggestion and have included this as a limitation. 

Although the transformation used is available in the literature [29], it may have limitations with regard to the analysis carried out because, ideally, some authors use an estimate of volumetric bone mineral apparent density, BMAD (grams per cm3), disclosed by Bachrach et al. [30] and adjusted by Zemel et al. [31]. In Bachrach's study the expression whole body BMC/height (cm) was calculated to adjust for whole body bone size, with statistical models according to sex and ethnicity. The study by Zemel et al. [31] used a complex predictive equation of height-for-age Z-score (HAZ). Due to the fact that our groups are paired, as previously indicated, with stature and height- for- age Z-score that did not differ statistically, the results are presented only in the Letter.

Values of BMD adjusted BMD Z-scores based on HAZ.

 Female (n=48) Male (n=36)

 MetS (-) (n=24) MetS (+) (n=24) MetS (-)

(n=18) MetS (+)

(n=18) 

 Mean ± SD P value Mean ± SD P value

Lumbar BMD 0.558 ± 1.00 0.547 ± 1.22 0.97 -0.511 ± 1.05 0.620 ± 0.88 0.049*

Total body BMD -0.665 ± 1.34 -0.377 ± 1.10 0.43 -0.740 ± 0.83 -0.628 ± 0.84 0.69

Subtotal BMD -0.684 ± 1.47 -0.223 ± 0.90 0.20 -0.601 ± 0.75 -0.474 ± 0.83 0.63

Lumbar Spine (BMD) for Age Z-Score adjusted for HAZ (height-for-age Z-score)

Total Body BMD for Age Z-Score adjusted for HAZ (height-for-age Z-score)

Total Body Less Head BMD for Age Z-Score adjusted for HAZ (height-for-age Z-score)

May add these references: 

We thank you for sending the references, they have all been added to the study.

Bachrach LK, Hastie T, Wang MC, Narasimhan B, Marcus R1999Bone mineral acquisition in healthy Asian, Hispanic, black, and Caucasian youth: a longitudinal study. J Clin Endocrinol Metab84:4702–4712.

Height Adjustment in Assessing Dual Energy X-Ray Absorptiometry Measurements of Bone Mass and Density in Children; Babette S. Zemel, Mary B. Leonard, Andrea Kelly, Joan M. Lappe,Vicente Gilsanz, Sharon Oberfield, Soroosh Mahboubi, John A. Shepherd,Thomas N. Hangartner, Margaret M. Frederick, Karen K. Winer,and Heidi J. Kalkwarf.

---

## [Editor Report · Decision Letter 3]

16 Jun 2021

Impact of metabolic syndrome and its components on bone remodeling in adolescents

PONE-D-20-24612R3

Dear Dr. Greenberg,

We’re pleased to inform you that your manuscript has been judged scientifically suitable for publication and will be formally accepted for publication once it meets all outstanding technical requirements.

Kind regards,

Benjamin Udoka Nwosu, MD

Academic Editor

PLOS ONE

Additional Editor Comments (optional):

Please rephrase the Conclusion as follows: 'Metabolic syndrome may be associated with reduced bone mineral density and biochemical markers of bone formation and resorption in adolescents with excess

weight.'
---

## [Editor Report · Acceptance letter]

22 Jun 2021

PONE-D-20-24612R3 

Impact of metabolic syndrome and its components on bone remodeling in adolescents 

Dear Dr. Goldberg:

I'm pleased to inform you that your manuscript has been deemed suitable for publication in PLOS ONE. Congratulations! Your manuscript is now with our production department. 

Kind regards, 

on behalf of

Dr. Benjamin Udoka Nwosu 

Academic Editor

PLOS ONE